# Single-cell transcriptomics uncovers a non-autonomous *Tbx1*-dependent genetic program controlling cardiac neural crest cell development

Christopher De Bono [1] ✉, Yang Liu [1], Alexander Ferrena [1,2], Aneesa Valentine[1], Deyou Zheng [1,3,4] & Bernice E. Morrow [1,5] ✉

Disruption of cardiac neural crest cells (CNCCs) results in congenital heart disease, yet we do not understand the cell fate dynamics as these cells differentiate to vascular smooth muscle cells. Here we performed single-cell RNA-sequencing of NCCs from the pharyngeal apparatus with the heart in control mouse embryos and when *Tbx1*, the gene for 22q11.2 deletion syndrome, is inactivated. We uncover three dynamic transitions of pharyngeal NCCs expressing *Tbx2* and *Tbx3* through differentiated CNCCs expressing cardiac transcription factors with smooth muscle genes. These transitions are altered non-autonomously by loss of *Tbx1*. Further, inactivation of *Tbx2* and *Tbx3* in early CNCCs results in aortic arch branching defects due to failed smooth muscle differentiation. Loss of *Tbx1* interrupts mesoderm to CNCC cell-cell communication with upregulation and premature activation of BMP signaling and reduced MAPK signaling, as well as alteration of other signaling, and failed dynamic transitions of CNCCs leading to disruption of aortic arch artery formation and cardiac outflow tract septation.

Neural crest cells (NCCs) are multipotent cells that migrate in three ordered streams from the rhombomeres in the neural tube to the pharyngeal apparatus, where they differentiate into many cell types[1]. The pharyngeal apparatus is a dynamic embryonic structure consisting of individual pharyngeal arches (PA), forming in a rostral to caudal manner from mouse embryonic day (E) 8 to E10.5. A subset of pharyngeal NCCs migrate through the caudal pharyngeal arches, PA3-6, and surround the pharyngeal arch arteries (PAAs), while others continue to migrate to the cardiac outflow tract (OFT), both differentiating to vascular smooth muscle cells[2].

The PAAs form in a similar, rostral to the caudal manner and are linked continuously with the cardiac OFT. The PAAs become remodeled to form the aortic arch and vascular branches that include the subclavian arteries. The fourth PAAs are particularly important in that the left fourth PAA is critical to form the aortic arch with the left subclavian artery and the right fourth PAA forms the branch to the right subclavian artery. Differentiation of CNCCs into smooth muscle cells of the PAAs is important for the remodeling of the PAA[2].

The OFT elongates by the addition of second heart field mesoderm cells and this is required for looping and septation of the OFT. It has been shown that CNCCs are required for the addition of second heart field mesoderm cells to expand the length of the OFT[3]. In addition, CNCCs are required for OFT septation by invading and allowing the endocardial cushions of the OFT to come into direct cell-cell

[1]Department of Genetics, Albert Einstein College of Medicine, Bronx, NY, USA. [2]Institute for Clinical and Translational Research, Albert Einstein College of Medicine, Bronx, NY, USA. [3]Department of Neurology, Albert Einstein College of Medicine, Bronx, NY, USA. [4]Department of Neuroscience, Albert Einstein College of Medicine, Bronx, NY, USA. [5]Departments of Obstetrics and Gynecology; and Pediatrics, Albert Einstein College of Medicine, Bronx, NY, USA. ✉e-mail: christopher.debono@einsteinmed.edu; bernice.morrow@einsteinmed.edu

contact, fuse, and form the aorticopulmonary septum[2,4,5]. Complete ablation of NCCs from PA3-6 results in interruption of the aortic arch and arterial branching defects, as well as persistent truncus arteriosus of the OFT[6]. These NCCs in PA3-6, are referred to as cardiac NCCs (CNCCs) based upon their position and known function in heart development as well as their differentiation to vascular smooth muscle. Understanding CNCC development is critical to determine the pathogenesis of human congenital heart defects such as those observed in 22q11.2 deletion syndrome (22q11.2DS) patients[7,8].

*TBX1*, encoding a T-box transcription factor, is the major gene for congenital heart disease in 22q11.2DS. Although 22q11.2DS is largely considered to be a neurocristopathy, *Tbx1* is not significantly expressed in CNCCs[9], but it is strongly expressed in adjacent cells in the pharyngeal apparatus, including the mesoderm. Global inactivation of *Tbx1* or conditional inactivation in the mesoderm using *Mesp1^Cre*[10] in the mouse results in neonatal lethality with a persistent truncus arteriosus[11–13], in part due to failed CNCC development[9]. Therefore, one of the main functions of *Tbx1* in the pharyngeal mesoderm is to signal to CNCCs to promote their development. In order to understand how CNCCs are affected non-autonomously in *Tbx1* mutant embryos, it is essential to define their transcriptional signatures and cardiac fate acquisition in the normal situation between E8.5 and E10.5, when *Tbx1* is expressed in the pharyngeal apparatus and when inactivated, on a single-cell level.

Previously, single-cell RNA-sequencing (scRNA-seq) of NCCs from early stages in the chick embryo identified expression of *Tgif1*, *Ets1*, and *Sox8* being important for early CNCC identity and fate decisions[14]. However, these were early migrating mesenchymal NCCs that also have the potential to contribute to the craniofacial skeleton and other cell types. Another seminal scRNA-seq study demonstrated that NCC fate choices are made by a series of sequential binary decisions in mouse embryos at E8.5-10.5[15] but did not focus on detailed steps of cardiac fate acquisition or investigate *Tbx1* function[15]. In another recent study using mouse scRNA-seq data, Chen and colleagues focus on CNCC differentiation from E10.5 to P (postnatal day) 7, once *Wnt1-Cre* derived NCCs have migrated to their final destinies in the heart[16]. However, this study did not include an analysis of the cardiac progenitor cells in the pharyngeal arches.

To uncover genetic signatures and dynamic transitions of CNCCs in the normal situation and when *Tbx1* is inactivated, we performed scRNA-seq of NCCs from control and *Tbx1* null mutant mouse embryos. We found that smooth muscle cell fate acquisition is in part dependent on two other T-box genes, *Tbx2* and *Tbx3*. When *Tbx1* is inactivated, we found reduced cell deployment (reduced presence of cells) along with the failure of dynamic progression of CNCC maturation. This is due to the disruption of cell-cell communication from mesodermal cells, resulting in the downregulation of MAPK signaling and upregulation and premature activation of the BMP pathway, as well as affecting other ligand–receptor interactions.

## Results

### Single-cell transcriptional profiling of NCCs in the pharyngeal apparatus

We performed scRNA-seq of the *Wnt1-Cre;ROSA-EGFP* genetic lineage[17,18] in the mouse pharyngeal apparatus at E8.5, E9.5 and E10.5 (Fig. 1a–f). These stages correspond to developmental time points when *Tbx1* is highly expressed in cell types adjacent to NCCs. At E8.5, the anterior half of the embryo was dissected and used for the scRNA-seq experiment because the embryo was too small to microdissect the pharyngeal apparatus with heart (Fig. 1a). At E9.5, the pharyngeal apparatus with heart was microdissected with the neural tube at the level of the pharyngeal apparatus, excluding the head and caudal half of the embryo in order to enrich for NCCs relevant to the PAAs and heart (Fig. 1b). At E10.5, the first arch, PA1 is large in size in proportion to the rest of the pharyngeal apparatus and contains NCCs that

contribute to the craniofacial skeleton. We excluded PA1 in order to enrich for NCCs relevant to cardiac development. At E10.5, arches two to six and the heart were included in the dissection (Fig. 1c). EGFP-positive NCCs were purified by FACS and the Chromium 10X platform was used to perform scRNA-seq and data from 36,721 NCCs of control embryos were obtained (Supplementary Table 1). Unsupervised clustering was performed using Seurat software[19] and individual clusters were identified (Fig. 1d–f and Supplementary Figs. 1–3).

Expression of *Sox10* and *Foxd3*, as well as expression of *Twist1*, *Prrx1*, *and Prrx2* were used to identify early migratory and mesenchymal NCCs, respectively[20,21]. We used *Hox* and *Dlx* (Homeodomain) genes to provide spatial anterior-posterior and proximal-distal context to different arches (PA2-6[22]). At E8.5, *Sox10* and *Twist1* show overlap in expression, while at E9.5, the expression became complementary, with a relative reduction of *Sox10* + NCCs and increased *Twist1* + NCCs in the expanded populations of mesenchymal NCCs (Fig. 1g, h). At E8.5 and E9.5, *Hoxa2* was expressed in PA2 and caudal arches, while *Hoxb3* was expressed only in PA3 and more caudal arches containing NCCs that will invade the OFT and surround the PAAs (Fig. 1d, e, g, h). Using the *Hox* genes as a guide, at E9.5, early migrating *Sox10* + NCCs of PA2 and PA3 were clustered together (cluster C4), suggesting that they have a similar transcriptional profile. At E10.5, the relative proportion of *Twist1*-expressing mesenchymal cells increased with respect to the reduction of *Sox10*-expressing cells (Fig. 1f, i). Further, at E10.5, *Hoxa2* was expressed within the mesenchymal cell populations from PA2-6, and *Hoxb3* was expressed in NCCs of PA3-6 (Fig. 1f, i). Additional marker genes are shown in Supplementary Fig. 1 (E8.5), Supplementary Fig. 2 (E9.5), and Supplementary Fig. 3 (E10.5) and listed in Supplementary Data 1 (E8.5), 2 (E9.5), and 3 (E10.5). Spatial localization was confirmed for expression of *Sox10*, *Hoxa2*, *Hoxb3*, and *Twist1* by wholemount RNAscope in situ hybridization (Fig. 1j–l). In addition, to anterior-posterior spatial localization of the cells, we identified their proximal-distal location in the PAs with *Dlx2*, *Dlx5*, and *Dlx6* (Supplementary Fig. 4). NCCs of the neural tube were identified based on the expression of *Zic1 and Zic2* genes and additional neural gene markers (Supplementary Figs. 1–3 and Supplementary Data 1–3).

### Identification of cardiac NCC gene signatures

Differentiated NCCs of the OFT and PAAs express smooth muscle genes such as smooth muscle actin, *Acta2*[23,24]. *Acta2* is a representative marker gene of progenitor smooth muscle cells, but it is not exclusive to smooth muscle cell types. Early markers of smooth muscle cells in addition to *Acta2* are *Tagln*, *Myl9*, *Myh9*, *Rgs5*, and *Cnn1*[25–27]. We used *Tagln* (Transgelin), *Cnn1* (Calponin), and *Rgs5* (Regulator of G Protein Signaling 5) as markers (Figs. 2, 3). CNN1 is a basic smooth muscle protein[28], RGS5 is expressed in vascular smooth muscle and pericytes[29] and TAGLN is ubiquitously expressed in vascular smooth muscle cells and is an early marker of smooth muscle differentiation[30]. This cell population also expresses additional markers of smooth muscle progenitors (Supplementary Data 3). By combining results from *Wnt1-Cre;GFP* lineage tracing experiments, immunofluorescence, and RNA-scope experiments of the early smooth muscle genes (cluster C14, Fig. 2a; C10, Fig. 3a; and Supplementary Figs. 2, 3), we refer to these cells as SM-CNCCs (smooth muscle expressing CNCCs). Furthermore, we performed co-immunostaining at E10.5 for *Wnt1-Cre;GFP*, ACTA2, and MF20 (myosin heavy chain antibody) to discriminate cardiac skeletal muscle cells, that also express ACTA2, from adjacent SM-CNCCs in the OFT. Supplementary Fig. 5 shows that MF20 is detected, as expected, in cardiac skeletal muscle cells but not in SM-CNCCs of the OFT.

To further delineate molecular signatures of CNCCs, we evaluated genes that were co-expressed in the SM-CNCCs (cluster C14, Fig. 2a; cluster C10, Fig. 3a). We identified known transcription factor genes involved in cardiac development, but not all of these are known to have a function in CNCCs development, including *Tbx2, Tbx3, Msx2,*

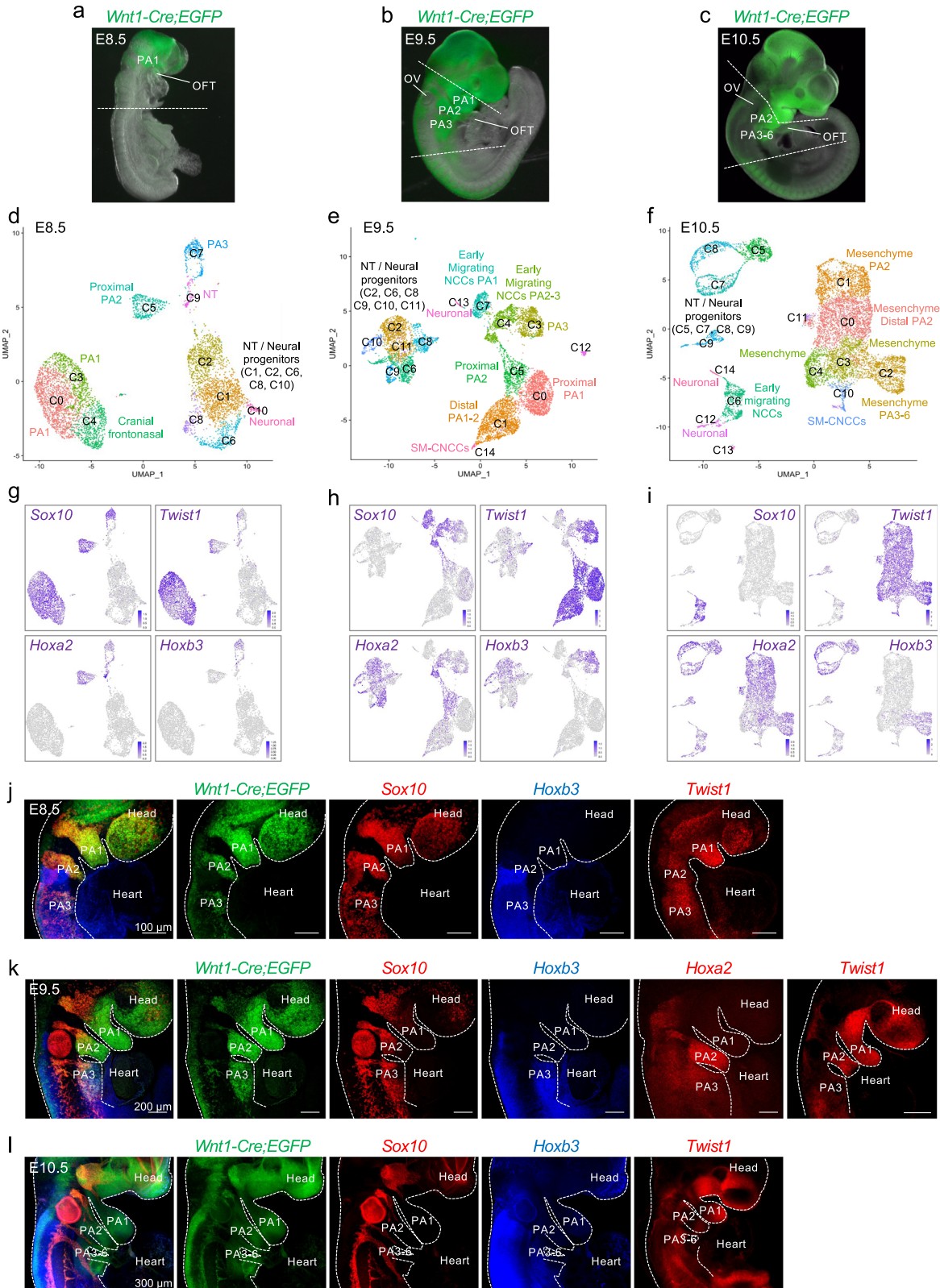

*Isl1*, *Gata3*, and *Hand2*, at E9.5 and E10.5 (Figs. 2a, 3a). These genes were not only expressed in SM-CNCCs of the OFT but also in other cell clusters, such as NCCs in the distal PA1-3 expressing *Dlx5* at E9.5 (C1, C3; Fig. 2a) and in PA3-6 at E10.5 (C2, C3, C4; Fig. 3a). At E9.5, we validated the expression of ACTA2 and TAGLN in SM-CNCCs in the distal part of the OFT (Fig. 2b, c) and co-expression of ISL1 in SM-CNCCs of which some expressed ACTA2 (Fig. 2b). Further, RNAscope

in situ analysis confirmed the expression of *Gata3*, *Isl1*, and *Msx2* in CNCCs within the OFT at E9.5 (Fig. 2d–f). At E10.5, *Isl1* and *Gata3* were expressed in CNCCs in the OFT and some also expressed ACTA2, *Cnn1*, *Rgs5*, and TAGLN (Fig. 3a–f) within the cardiac cushions of the distal OFT and in the mesenchyme of the dorsal aortic sac wall and aortic sac protrusion. *Gata3* was expressed in a larger domain of the OFT than *Isl1* (Fig. 3c). When taken together, at this time point, CNCCs of the OFT

**Fig. 1 | Single-cell RNA-seq of NCCs from mouse embryos at E8.5-E10.5 reveals transcriptional heterogeneity within the pharyngeal region. a–c** *Wnt1-Cre;R-OSA-EGFP* genetic lineage tracing shows the distribution of NCCs within the pharyngeal region and outflow tract of E8.5 (**a**), E9.5 (**b**), and E10.5 (**c**) embryos. The region rostral to the white dotted line of the embryo at E8.5 (**a**) and the pharyngeal region between the dotted lines in embryos at E9.5 and E10.5, including the heart were microdissected and EGFP-positive NCCs were used for scRNA-seq. **d–f** Seurat UMAP (Uniform Manifold Approximation and Projection) plots with cluster annotations of scRNA-seq data of NCCs at E8.5 (**d**), E9.5 (**e**), and E10.5 (**f**). **g–i** Expression of genes at E8.5 (**g**), E9.5 (**h**), and E10.5 (**i**) with highest expression in blue and lowest expression in gray. **j–l** Wholemount RNAscope in situ hybridization of *Wnt1-Cre;ROSA-EGFP* embryos (*n* = 3) at E8.5 (**j**), E9.5 (**k**), and E10.5 (**l**) with probes for *Sox10*, *Hoxb3*, and *Egfp*, together and separated (colors are indicated above embryos). *Twist1* expression was analyzed at E8.5, E9.5, and E10.5 and *Hoxa2* was examined at E9.5 as indicated. PA pharyngeal arch, OFT outflow tract, OV otic vesicle, SM-CNCCs smooth muscle CNCCs. Scale bar: 100 µm in **j**, 200 µm in **k**, and 300 µm in **l**.

derived from the *Wnt1-Cre;GFP* genetic lineage and expressing *Acta2*, *Tagln*, *Cnn1*, *Rgs5*, *Isl1*, and *Gata3* are smooth muscle progenitors.

*Tbx2 and Tbx3* were widely expressed in pharyngeal NCCs at E9.5 (Fig. 2a), but their expression was restricted to cell clusters comprising PA3-6 at E10.5 (Fig. 3a). *Tbx2* and *Tbx3* were expressed immediately lateral and dorsal to *Isl1* and *Gata3* expressing CNCCs in embryos at E10.5 as determined by RNAscope analysis (Fig. 3g–i). A *Tbx2* and *Gata3* co-expression domain was present between *Tbx2* and *Gata3* expressing CNCCs (Fig. 3g). Together, this suggests that CNCCs downregulate *Tbx2* and *Tbx3* expression and activate *Isl1* and *Gata3* expression when entering the OFT. In addition, *Tbx2* and *Tbx3*, but not *Isl1* and *Gata3*, were expressed in NCCs surrounding the PAAs that are differentiating to smooth muscle, expressing ACTA2 and TAGLN at E10.5 (Fig. 3f, h, i). Both ISL1 and TBX2 proteins were expressed in smooth muscle cells of the OFT and PAAs, respectively (ACTA2 or TAGLN; Fig. 3b, f). Expression of *Tbx2*, *Tbx3*, *Isl1*, *Gata3*, *Tagln*, and *Acta2* in NCCs at E10.5 is illustrated in Fig. 3j. A subset of pharyngeal NCCs, expressing *Hoxb3* that serves as a marker for PA3-6 as well as *Tbx2* and *Tbx3*, will form the CNCCs (cluster C2) and we refer to these cells as cardiopharyngeal NCCs (CP-NCCs; Fig. 3j).

Therefore, CNCCs can be subdivided into four populations based on position and expression of cardiac or smooth muscle genes, referred to as CP-NCCs (contained in clusters C1, C3, C5, E9.5; C2, E10.5), PAA-CNCCs of PAAs expressing *Acta2, Tagln, Tbx2*, and *Tbx3* (contained in C10, E10.5), OFT-CNCCs of the OFT expressing *Isl1* and *Gata3* (contained in Clusters C1 and C3 E9.5; C3 and C4, E10.5) and SM-CNCCs of the OFT that express *Acta2* and *Tagln* (Clusters C14, E9.5; C10, E10.5; Fig. 3j).

We noted earlier that some CNCCs were located in clusters from PA1 and PA2 at E9.5 (C1; Fig. 2a), that are not typically considered to harbor CNCCs. Consistent with this, at E9.5 (20 somites), we found that the OFT was connected to PA2, and CNCCs from PA2 were entering the OFT (Fig. 2g, left). At late E9.5 (24 somites), the OFT was located between PA2 and PA3, and CNCCs from PA2 and PA3 were entering the OFT (Fig. 2g, right). These data are consistent with evidence from a previous report[31], that NCCs from anterior arches also contribute to the developing heart.

**Cardiac NCC fate dynamics that drive differentiation to smooth muscle cells**

To uncover CNCC fate dynamics at E9.5, we used CellRank software[32] (Fig. 2h, i). We discovered genes that were progressively activated during the transition from pharyngeal NCCs to SM-CNCCs, which are candidate cardiac lineage driver genes (Fig. 2j, k and Supplementary Data 4 for the full list of genes). Our analysis indicates that CNCCs progressively activate *Tbx2, Tbx3, Msx2, Hand2, Gata3, and Isl1* expression during their commitment towards SM-CNCCs at E9.5 (Fig. 2j, k).

To understand how NCCs progress at E10.5, when there are more SM-CNCCs in the pharyngeal arches, we used CellRank software and generated PAGA (partition-based graph abstract) plots (Fig. 4a–c). The cell fate probability map from CellRank identified cells with a high potential to differentiate to smooth muscle fates (from cluster C2 and C3 to C10; Fig. 4c). The PAGA plots further indicated that some pharyngeal NCCs (cluster C2) are CP-NCCs and they transition to OFT-CNCCs (cluster C3) that then transition to SM-CNCCs (cluster C10; blue color fraction in the pie chart; Fig. 4c). This data also indicates that a small fraction of CP-NCCs may directly differentiate to smooth muscle cells (blue fraction in the pie chart in C2 indicates a high probability of a fraction of cells in C2 to transition directly to C10), in agreement with *Tbx2* and *Tbx3* expression in PAA-CNCCs at E10.5 (Fig. 3f, h, i). We next identified genes whose expression correlates with SM-CNCC fate acquisition (Fig. 4d, e and see Supplementary Data 5 for a full list of genes at E10.5).

We mined the literature focusing on genes encoding transcription factors and signaling molecules, which traditionally have roles in cardiac development and modulating cell fate at E9.5 and E10.5 (Figs. 2j, 4d). Representative genes, with a focus on transcription factors, signaling genes, and early smooth muscle markers, were ordered according to their expression peak in pseudotime (Figs. 2j, 4d). While some are known to function in NCCs for heart development (*Sema3c, Msx2, Hand1, Hand2, Gata4, Gata6, Nrp1, Prdm6*, and *Smad7*), we found others that are newly recognized that may be required in NCCs for heart development, such as *Tbx2* and *Tbx3* (stars; Figs. 2j, 4d and Supplementary Fig. 6).

We generated lineage driver gene sets at E10.5 by dividing the genes in the fate probabilities heatmap from *Bmp4* to *Gata6*, least to most differentiated to SM-CNCCs, amounting to approximately 900 genes, to four groups of equal size (Supplementary Data 5). Next, we performed a Gene Ontology (GO) enrichment analysis of each set using ToppGene Suite[33] to understand the function of the genes in each group (Fig. 4f and Supplementary Data 6). Our analysis indicates that the initially activated genes of pharyngeal NCCs that include some CP-NCCs, are associated with general embryonic and mesenchyme development processes (e.g., *Hox, Dlx*, and *Six2* genes). Then cell division (cell cycle) genes are highly expressed, consistent with the expansion of pharyngeal NCCs during development[34], together with cardiac development genes. Finally, genes important in cardiac development and actin-filament processes (*Hand1, Gata3/4/5/6, Isl1*, and *Acta2*) become strongly expressed (Fig. 4f). In addition, our functional enrichment analysis of the fourth, most differentiated set, identified genes associated with congenital heart diseases (*Gata4/6, Hand1/2, Tbx20, Nrp1*, and *Acta2*) such as tetralogy of Fallot, double outlet right ventricle and ventricular septal defects in the most differentiated cells (Supplementary Data 6), which supports the importance of the genetic program of CNCCs in OFT formation and disease.

Furthermore, we integrated our scRNA-seq data from control embryos at E8.5, E9.5, and E10.5 and investigated the cardiac cell fate trajectory (Supplementary Fig. 7) using the same approach as for individual data analysis at E9.5 and E10.5. Cardiac fate trajectory analysis of the integrated data provides confirmatory information of our results from individual data analysis as progressive activation of the same specific genes during cardiac fate acquisition were found. Thus, here we identified a specific CNCC transcriptomic signature at E9.5-10.5 and revealed that cell fate acquisition to smooth muscle cells requires a multistep specification process. We additionally identified transcription factor and signaling genes such as *Dkk1, Gata3, Foxf1, Mef2c, Isl1, Isl2, Tbx2, Tbx3, Rgs5, Bambi*, and *Smad6*, among others, which have not yet been considered as CNCC transition markers (Supplementary Data 4, 5).

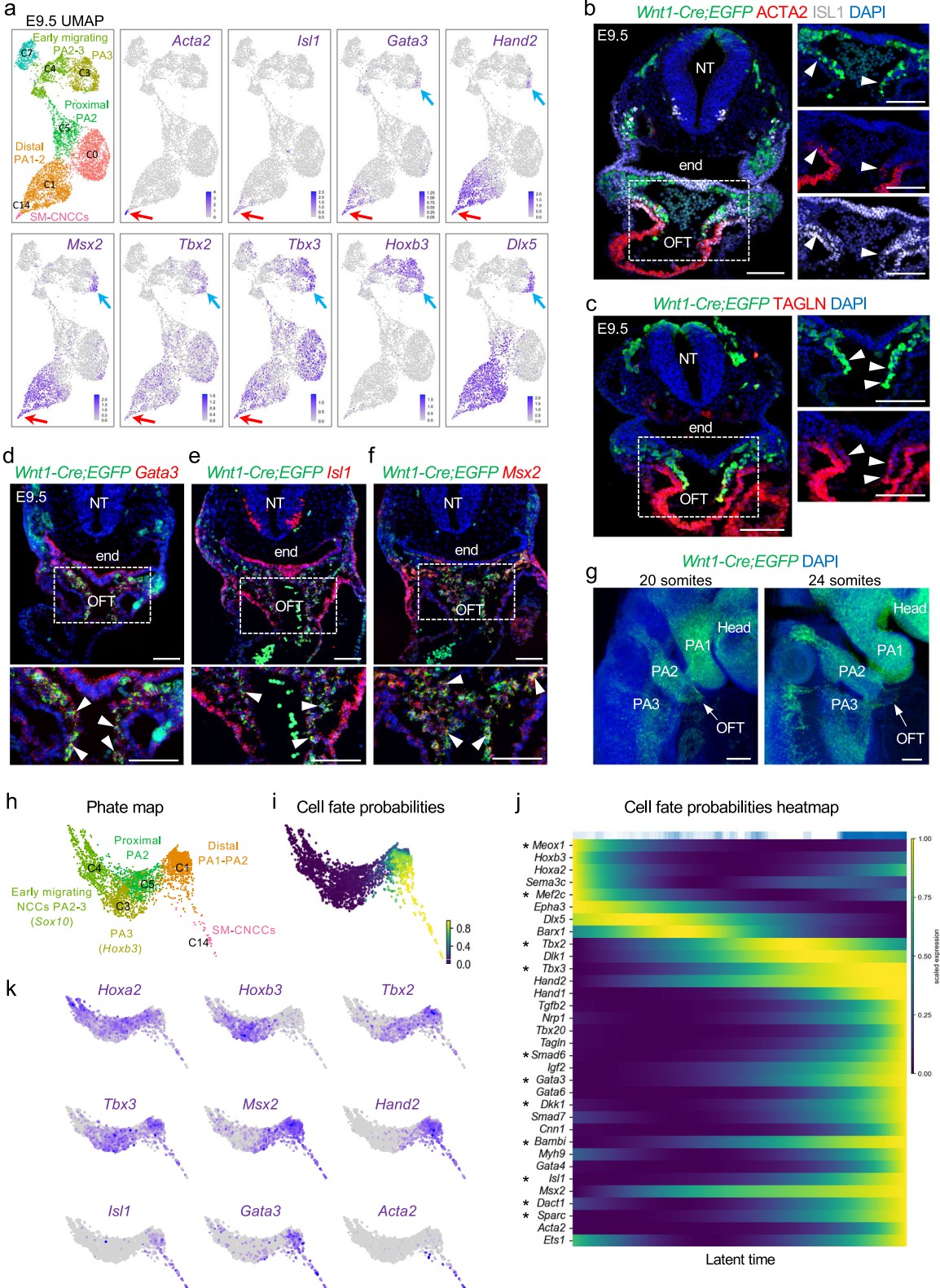

**Tbx2 and Tbx3 are required in cardiac NCCs for aortic arch branching**

*Tbx2* and *Tbx3* are expressed in multiple tissue types within the phar-yngeal apparatus and global inactivation of both genes lead to early embryonic lethality with severe cardiac defects[35–38]. To understand the requirement of *Tbx2* and *Tbx3* in NCCs, we generated *Wnt1-Cre/+; Tbx2^{f/f};Tbx3^{f/f}* double conditional mutant embryos (*Tbx2/3* cKO).

We performed intracardiac ink injection, arterial branching analysis, and histological analysis at E15.5 (Fig. 5a–l) and found that 38.5% of *Tbx2/3* cKO embryos had an aberrant retro-esophageal right subclavian artery (ARSA), 25% of *Tbx2/3* cKO embryos had hypoplasia of the aortic arch but no intracardiac defects (Fig. 5a–l, m). No defects were identi-fied in *Wnt1-Cre/+;Tbx2^{f/f};Tbx3^{f/+}* nor in *Wnt1-Cre/+;Tbx2^{f/+};Tbx3^{f/f}*, in which three of four alleles were inactivated in embryos.

**Fig. 2 | Transcriptional dynamics of cardiac NCCs at E9.5. a** UMAP plots of scRNA-seq data with genes that mark CNCCs identified by expression of *Acta2*, *Isl1*, *Gata3*, *Hand2*, *Msx2*, *Tbx2*, and *Tbx3* with respect to *Hoxb3* (PA3) and *Dlx5* (distal PA1, 2, 3) expression (red arrows). Blue arrows show the expression of *Gata3*, *Hand2*, *Msx2*, *Tbx2*, and *Tbx3* in the distal part of PA3. **b, c** Immunostaining on traverse sections through *Wnt1-Cre;ROSA-EGFP* embryos showing *Cre*-activated EGFP (green), ACTA2, TAGLN, and ISL1 protein expression. Nuclei (blue) are labeled with DAPI. White arrowheads indicate CNCCs expressing ACTA2 and ISL1 ($n = 3$) (**b**) and CNCCs expressing TAGLN ($n = 3$) (**c**) in the OFT. **d–f** RNAscope analysis of *Wnt1-Cre;ROSA-EGFP* embryos for *Egfp*, *Gata3* (**d**) ($n = 3$), *Isl1* (**e**) ($n = 3$), and *Msx2* (**f**) ($n = 4$) expression. Nuclei (blue) are labeled with DAPI. Arrowheads indicate the expression of *Gata3*, *Isl1*, and *Msx2* in CNCCs within the OFT. **g** Wholemount RNAscope analysis of *Wnt1-Cre;ROSA-EGFP* embryos at 20 and 24 somites where the position of the OFT is indicated (white arrowhead). **h** PHATE (potential of heat-diffusion for affinity-based transition embedding) map of NCCs in clusters C1, C3, C4, C5, and C14 using Louvain clustering. **i** PHATE map colored by cell fate probabilities, showing how each cell is likely to transition to SM-CNCCs as defined by CellRank software (yellow represents high cell fate probabilities). **j** Heatmap from CellRank showing the expression of marker genes whose expression correlates with cardiac fate probabilities as latent time, with cells order by fate probabilities as latent time (see Supplementary Data 4 for a full list of genes), Asterisk (*) indicates newly identified genes connected to CNCCs. **k** PHATE maps with an expression of marker genes of CNCCs at different states of differentiation towards smooth muscle *Acta2* expressing cells. NT neural tube, end endoderm, OFT outflow tract, PA pharyngeal. Scale bars: 100 µm.

The right subclavian artery and aortic arch are formed from the right and left 4th PAA, respectively. By immunostaining on coronal sections of *Wnt1-Cre/+;Tbx2^f/f^;Tbx3^f/f^;ROSA-EGFP^f/+^* embryos at E11.5 using GFP and ACTA2 antibodies, we found that NCCs contributed normally to the right and left fourth PAAs but partially or completely failed to differentiate into smooth muscle cells (Fig. 5n–q). The failure of differentiation ranges from almost no expression of ACTA2 (Fig. 5o) to a low level of expression (Fig. 5p, q). This variation might explain the partial penetrance of defects in mutant embryos. We qualitatively analyzed control versus *Wnt1-Cre/+;Tbx2^f/f^;Tbx3^f/f^* embryos at E15.5 to compare to the thickness of the muscular wall of the RSA (control and *Tbx2/3* cKO without ARSA) and with ARSA (Supplementary Fig. 8). We found reduced thickness and reduced number of muscle layers in the ARSA of *Wnt1-Cre/+;Tbx2^f/f^;Tbx3^f/f^* embryos, suggesting an absence of the smooth muscle layer, consequently to a possible failure of smooth muscle differentiation. *Bmp4* and *Foxf1* have been identified as regulators of smooth muscle cell differentiation in other organs[39]. We found that *Bmp4* and *Foxf1* expression was activated temporally after *Tbx2* and *Tbx3* expression during cardiac fate acquisition (Fig. 4d).

### Disruption of cardiac NCCs by loss of *Tbx1*

In *Tbx1* null mutant embryos, the caudal pharyngeal apparatus is hypoplastic and unsegmented at E9.5 and E10.5 due in part to the failed deployment of NCCs[9]. Further, CNCCs fail to enter the shortened cardiac OFT, leading to a persistent truncus arteriosus later in development[9]. We found that *Tbx1* was not noticeably expressed in NCCs (Supplementary Fig. 9) and its conditional deletion in NCCs using *Wnt1-Cre* did not lead to cardiac defects (Supplementary Fig. 10).

To understand how the absence of *Tbx1* non-autonomously affects the development of CNCCs, we performed scRNA-seq of NCCs isolated from the microdissected pharyngeal region plus the heart of *Tbx1* null mutant embryos at E9.5 (Fig. 6a). We obtained sequencing data from 11,301 NCCs (Supplementary Table 1) and integrated scRNA-seq data from control and *Tbx1* null embryos using RISC (Robust Integration of scRNA-seq) software[40] (Fig. 6b). Even though there were visibly fewer NCCs in the pharyngeal apparatus (Fig. 6a), surprisingly, there were no missing cell clusters in *Tbx1* null embryos (Fig. 6b, c). We confirmed that most of the cells in each cluster of the integrated control and *Tbx1* null scRNA-seq dataset (shown in Fig. 6b) were also clustered together in the individual datasets (Fig. 1e and Supplementary Fig. 11). This indicates that although affected, the general transcriptional profiles of the CNCCs are still conserved between control and *Tbx1* null embryos. We then compared the proportion of cells in each cluster among the total number of NCCs in each dataset. As expected, there was a reduction in the relative proportion of NCCs in *Tbx1* null embryos as compared to controls, as shown in Fig. 6d, in PA3 (C4,1.5 fold), in proximal PA2 (C8, 1.4-fold), in early migrating NCCs in PA2 and PA3 (C9, 1.2-fold), and in early migrating NCCs in PA1 (C5, 1.4-fold). In addition, there was an increase in the relative proportion of NCCs in the distal part of PA1 and PA2 (C3; 1.6-fold; Fig. 6d). The increase in the proportion of NCCs is consistent with previous data

that cells from PA2 abnormally migrate to PA1 in *Tbx1* null embryos at this stage[9].

### Altered BMP and MAPK signaling pathways in the absence of *Tbx1*

We examined the data to identify differentially expressed genes (DEGs) in mutant versus control embryos at E9.5 (Supplementary Data 7). One of the most notable changes in *Tbx1* null embryos was an increase in the expression of genes that act downstream of BMP signaling in proximal PA2 and PA3 (clusters C4, C8; Fig. 6e–g). This includes increased expression of *Msx2*, *Msx1*, *Bambi*, *Dkk1*, *Smad7*, *Id2*, and *Id3*. Our analysis also revealed a downregulation of the expression of genes in the MAP kinase (mitogen-activated protein kinase) signaling pathway, including *Spry2*, *Spry4*, *Myc*, *Foxo1*, and *Lyn* (Fig. 6e–g). Signaling by BMP[41] and growth factors activating the MAPK pathway[42,43] are two signaling pathways known to be critical in NCCs for cardiac development. The increase in expression of *Msx2* (downstream in the BMP pathway), as well as reduced expression of Spry1 (MAPK pathway) in scRNA-seq data of *Tbx1* null embryos, was observed in feature plots (Fig. 7a). We confirmed reduced MAPK signaling in pharyngeal NCCs of *Tbx1* null embryos at E9.5 by performing wholemount immunostaining for P-ERK (phospho-ERK; Fig. 7b). Expression of BMP downstream genes, *Msx2* and *Bambi*, were expanded dorsally in *Tbx1* null mutant embryos by wholemount RNAscope in situ and 3D reconstruction (Fig. 7c). These results were confirmed by RNAscope assays on traverse sections of control and *Tbx1* null embryos (Fig. 7d). Normally, BMP signaling is most active in the ventral part of the pharyngeal apparatus near the OFT. We found an increase and ectopic expression of P-SMAD1/5/9 in the dorsal pharyngeal region, marking an increase in BMP signaling in NCCs in *Tbx1* null embryos (PA2, PA3; Fig. 7e). NCCs migrate from the dorsal to ventral parts of the pharyngeal apparatus. Our results indicate that the BMP pathway is prematurely activated during their migration in a more dorsal region of the pharyngeal apparatus compared to control embryos. These data suggest that altered BMP and MAPK signaling might affect NCC development in *Tbx1* null embryos. A schematic representation of expanded and premature BMP signaling activity and reduced NCCs migrating to the shortened OFT in *Tbx1* null embryos at E9.5 is shown in Fig. 7f.

### Cell-cell communication from the mesoderm to NCCs is disrupted in the absence of *Tbx1*

During the formation of the heart, NCCs receive critical signaling from adjacent mesodermal cells (Fig. 8a). We investigated cell-cell communication and how it is disrupted in the absence of *Tbx1* at single-cell resolution using CellChat software[44].

Inactivation of *Tbx1* in the mesoderm results in similar pharyngeal hypoplasia and altered NCC distribution as in global null embryos, implicating the pharyngeal mesoderm as being critical to signal to NCCs[45]. To identify *Tbx1*-dependent signals from the mesoderm to NCCs, we investigated existing scRNA-seq data from *Mesp1^Cre^* control and *Tbx1* conditional null embryos at E9.5[46]. The scRNA-seq from two

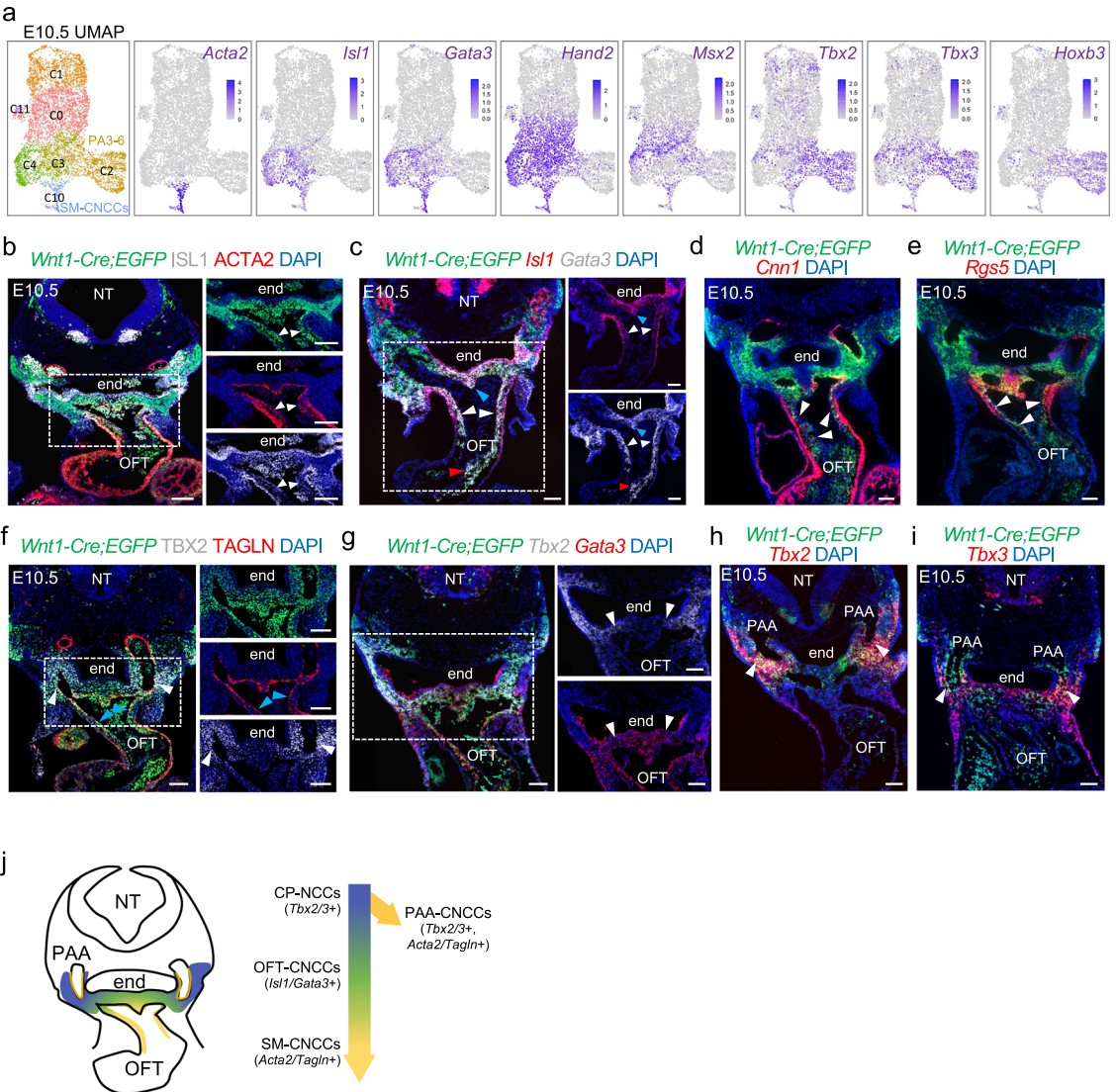

**Fig. 3 | Transcriptional dynamics of cardiac NCCs at E10.5. a** Seurat UMAP plots of scRNA-seq data from NCC populations with the expression of marker genes. **b** Immunostaining on traverse sections of *Wnt1-Cre;ROSA-EGFP* embryos (*n* = 3) showing GFP, ISL1, and ACTA2 expression. Nuclei stained with DAPI are in blue. ISL1 is expressed in smooth muscle cells of the OFT cushions (arrowheads). **c–e** RNAscope in situ hybridization sections of *Wnt1-Cre;ROSA-EGFP* embryos for *Egfp, Isl1* and *Gata*3 (**c**) (*n* = 5), *Egfp, Cnn1* (**d**) (*n* = 3), *Egfp* and *Rgs5* (**e**) (*n* = 3) expression. Nuclei stained with DAPI are in blue. *Isl1* and *Gata3* are expressed in NCCs within the OFT (white arrowheads in **c**) and at the level of the dorsal aortic sac wall and aortic sac protrusion (blue arrowheads in **c**). *Gata3* but not *Isl1* is expressed in the more proximal part of OFT (red arrowheads in **c**). *Cnn1* and *Rgs5* are expressed in smooth muscle cells of the OFT (white arrowheads in **d** and **e**).

**f** Immunostaining on sections of *Wnt1-Cre;ROSA-EGFP* embryos (*n* = 3) showing GFP, TBX2, and TAGLN expression. Expression of TAGLN is in smooth muscle cells of the OFT (blue arrowheads) and PAAs expressing TBX2 (white arrowheads). **g–i** RNAscope in situ hybridization on traverse sections of *Wnt1-Cre;ROSA-EGFP* embryos for *Egfp, Tbx2* (*n* = 5) and *Gata3* (*n* = 4) (**g**), *Egfp* and *Tbx2* (**h**) (*n* = 5), and *Egfp* and *Tbx3* (**i**) (*n* = 5) expression. Nuclei stained with DAPI are in blue. *Tbx2* and *Gata3* expression overlaps in the dorsal wall of the OFT (white arrowheads in **g**). *Tbx2* and *Tbx3* are expressed in the mesenchyme dorsal to the aorta surrounding the PAAs at the level of the PA3-6 (white arrowheads in **h** and **i**). **j** Schematic representation of a transverse section at the level of the OFT summarizing *Tbx2, Tbx3, Isl1, Gata3, Acta2,* and *Tagln* expression in CNCCs at E10.5. NT neural tube, end endoderm, PAA pharyngeal arch artery. Scale bars: 100 μm.

replicates of *Mesp1^Cre* control and *Tbx1* conditional null embryos were integrated using RISC software as described[46]. We focused on meso-dermal subpopulations, expressing *Tbx1*, that are adjacent to the NCCs, including the anterior and posterior second heart field (aSHF; pSHF)[47,48]. We also included a critical *Tbx1*-dependent multilineage progenitor population (MLP) in the pharyngeal mesoderm required for cell fate progression to the aSHF and pSHF[46]. We examined sig-naling to NCCs in clusters corresponding to migrating NCCs of the future PA2-3 (C9), distal part of PA1-2 (C3), mesenchyme of PA2 (C8) and PA3 (C4), and CNCCs of the OFT (C12) in integrated scRNA-seq data from control and *Tbx1* null embryos at E9.5 (Figs. 6, 8b). Repre-sentative results of ligand–receptor pairs altered when *Tbx1* is

inactivated are shown in Fig. 8b, and the complete set of pairs are in Supplementary Fig. 12.

Affected ligands in the mesoderm include *Wnt5a, Wnt2, Sema3c, Pdgfa Nrg1, Fgf8, Fgf10, Bmp4,* and *Edn3* and others. To validate rela-tionships, we analyzed integrated *Mesp1^Cre* data (Fig. 8c; MLPs-C8, aSHF-C10, and pSHF-C1 + C12). *Isl1*, is a critical gene required for OFT development[49], and it is expressed in the MLPs, aSHF, and pSHF (Fig. 8d). We examined expression changes in feature plots of *Wnt5a, Wnt2, Sema3c, Pdgfa, Nrg1, Fgf10,* and *Edn3* as well as other genes (Fig. 8e–m). These genes were altered in expression in the cell types specified and, in the direction of altered signaling (decreased or increased in the mutant embryos), as indicated in Fig. 8b.

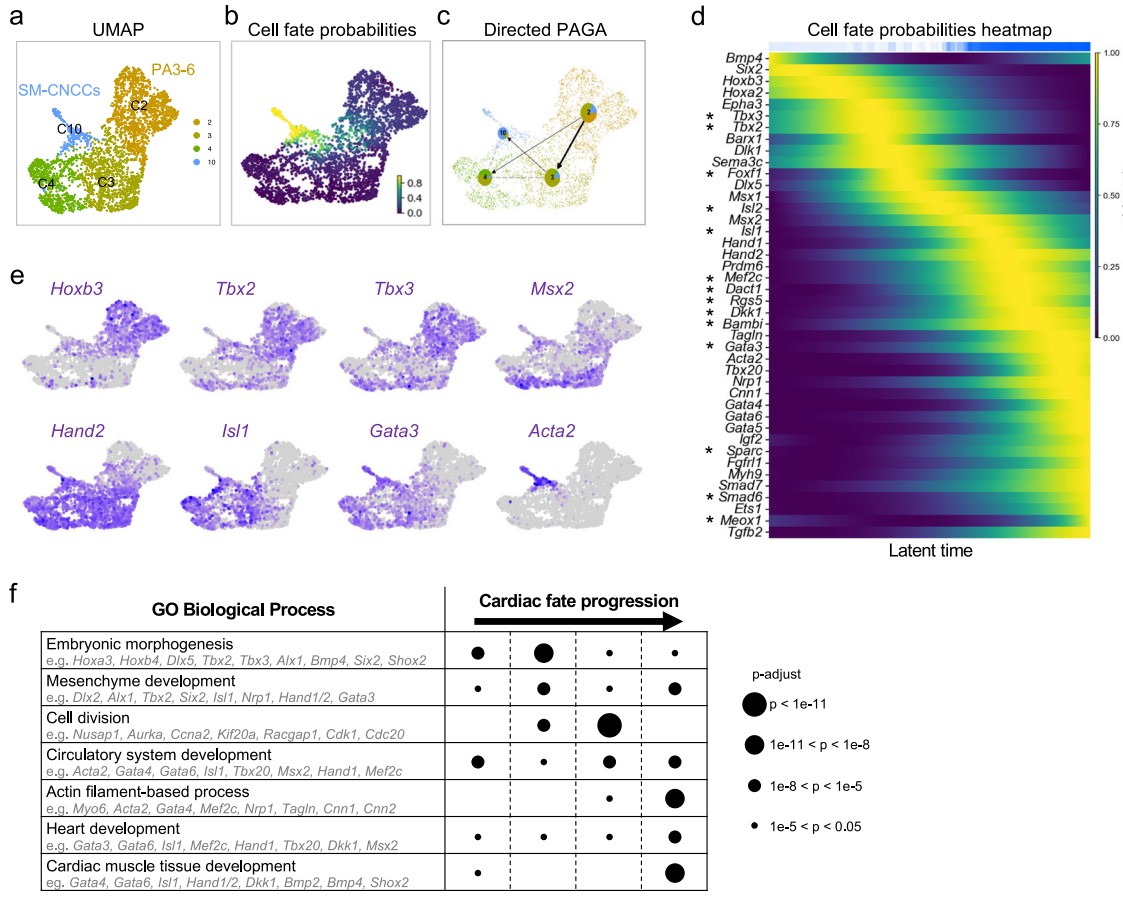

**Fig. 4 | Identification of transition genes and their function at E10.5. a** CellRank UMAP plot directed by RNA velocity and cell-cell similarity for clusters C2, C3, C4, and C10 from the feature plot in Fig. 3a. **b** UMAP plot for cell fate probabilities of CNCCs differentiating towards smooth muscle cells. **c** Directed PAGA plot of NCCs. Pie charts show summarized fate probabilities of individual clusters, with blue representing the proportion of cells in each cluster with high probability to become smooth muscle CNCCs of a cluster, C10. **d** Heatmap from CellRank showing the expression of selected genes whose expression correlates with transitions of CNCCs fate probabilities, with cells ordered by smooth muscle fate probabilities as latent time (see Supplementary Data 5 for a full list of genes), Asterisk (*) marks newly identified genes connected to CNCCs. **e** UMAP plots showing expression of marker genes in CNCCs. **f** GO enrichment analysis of four groups of genes between *Bmp4* and *Gata6* (defined by their pseudotime; Supplementary Data 6). Example of genes for each selected GO: biological processes are provided. The size of the dots indicates the adjusted *p* value by Bonferroni and Hochberg correction (Hypergeometric test); adjusted *p* value are indicated in Supplementary Data 6.

Reduced expression of *Wnt5a*[50,51], *Fgf8*[46,52,53], *Fgf10*[54,55], *Sema3c*[56], and *Nrg1*[46] ligands and increase of *Wnt2*[57] are consistent with previous in vivo studies of *Tbx1* mutant embryos; however, these were not known with respect to cell-cell communication to NCCs. With these data, we show, on a single-cell level, that signaling to NCCs is altered in *Tbx1* mutant embryos. We also identified ligand–receptor pairs that were not previously connected to NCCs or *Tbx1*. Two ligand genes, *Edn3* and *Pdgfa* were identified in our cell-cell communication analysis (Figs. 8h, k). *Edn3* encodes an endothelin ligand important in cell migration. *Pdgfa* encodes a growth factor regulating cell survival, proliferation, and migration and PDGF signaling is required in NCC development[58]. To our knowledge, there is no report of *Edn3* and *Pdgfa* having been investigated in relation to *Tbx1*.

We investigated CellChat results for the BMP and MAPK pathways that were altered in NCCs when *Tbx1* was inactivated. An abnormal increase of the BMP signaling from the mesoderm to NCCs in *Tbx1* mutant embryos was found through *Bmp4/5/7* ligands (Fig. 8b). We performed RNAscope in situ hybridization for expression of *Bmp4* and *Bmp7* (Fig. 8n and Supplementary Fig. 13). *Bmp4* is expressed in the pharyngeal mesoderm and *Bmp7* is expressed in the pharyngeal mesoderm and endoderm adjacent to the NCCs. Expression of these two ligands were maintained in *Tbx1* null embryos as compared to control embryos at E9.5 (Supplementary Fig. 13 and Fig. 8l, m).

*Fgf8* and *Fgf10* are ligands in FGF signaling that act through the MAPK pathway and it is well known that they are reduced in expression in *Tbx1* mutant embryos[46,52–55] (Fig. 8j). Similarly, we found that *Fgf8* and *Fgf10* were reduced in expression in the mesoderm and signaling to NCCs was altered (Fig. 8b). It is known that FGF signals to NCCs during embryonic development[42,59,60]. Therefore, it is possible that the reduction of FGF and/or increase in BMP, along with changes in other ligands in the adjacent mesoderm, could lead to their failed progression and cardiac contribution.

### Failed cardiac cell fate progression of NCCs in the absence of *Tbx1* at E10.5

To further understand how the contribution of NCCs to the OFT is altered in the absence of *Tbx1*, we performed scRNA-seq of NCCs in *Tbx1* null embryos at E10.5, when the caudal pharyngeal apparatus is hypoplastic (Fig. 9a). We integrated scRNA-seq data from two control replicates (21,561 cells) and two *Tbx1* null replicates (17,840 cells) using RISC software (Fig. 9b). We confirmed that most of the cells in each cluster of the integrated control and *Tbx1* null scRNA-seq dataset (shown in Fig. 9b) were also clustered together in the individual datasets (Fig. 1f and Supplementary Fig. 14). The integrated datasets clearly show a strong reduction in the number of NCCs in most clusters in *Tbx1* null embryos including OFT-CNCCs (*Isl1/Gata3*+; C10) and SM-

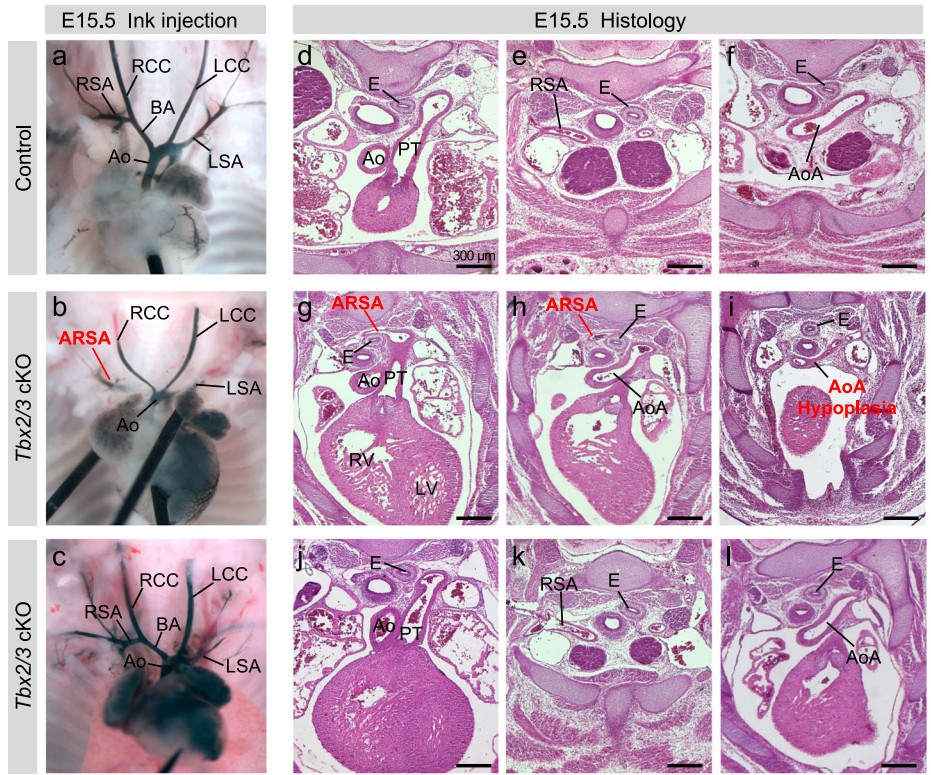

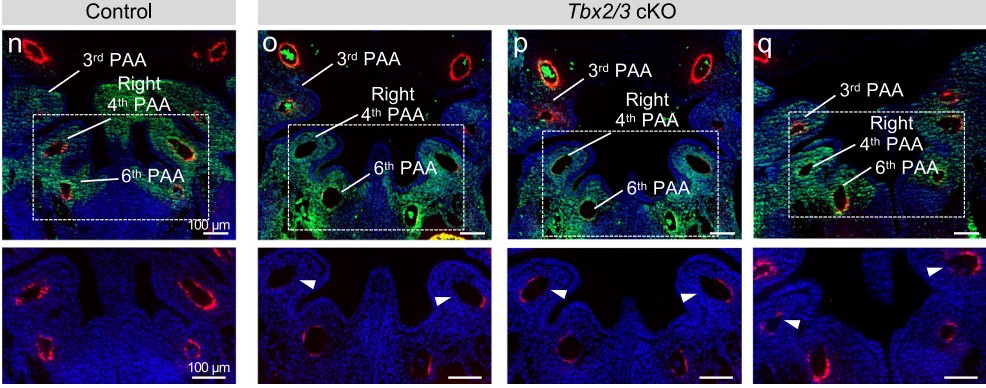

| Genotypes | Normal RSA | ARSA | Normal aortic arch | Aortic arch hypoplasia |
|---|---|---|---|---|
| *Tbx2^{f/f};Tbx3^{f/f}* <br> *Tbx2^{f/+};Tbx3^{f/f}*  (Control) <br> *Tbx2^{f/f};Tbx3^{f/+}* | 6/6 (100%) | 0 | 4/4 (100%) | 0 |
| *Wnt1-Cre/+;Tbx2^{f/+};Tbx3^{f/+}* | 3/3 (100%) | 0 | 2/2 (100%) | 0 |
| *Wnt1-Cre/+;Tbx2^{f/f};Tbx3^{f/+}* | 5/5 (100%) | 0 | 3/3 (100%) | 0 |
| *Wnt1-Cre/+;Tbx2^{f/+};Tbx3^{f/f}* | 7/7 (100%) | 0 | 6/6 (100%) | 0 |
| *Wnt1-Cre/+;Tbx2^{f/f};Tbx3^{f/f}*  (Tbx2/3 cKO) | 8/13 (61.5%) | 5/13 (38.5%) | 6/8 (75%) | 2/8 (25%) |

CNCCs (*Acta2*+; C13) as shown in Fig. 9c, except that the relative proportion of pharyngeal NCCs that contain CP-NCCs (*Tbx2/Tbx3*+; C4) was not significantly changed, after considering the total cells being sequenced (Fig. 9d).

Our analysis showed that the cell fate probabilities point from pharyngeal NCCs containing CP-NCCs (C4) and OFT-CNCCs (C10) toward SM-CNCCs (C13) as shown in Fig. 9e. We generated a list of

DEGs in clusters C4 and C10, between control and mutant embryos at E10.5 (Supplementary Data 8). *Msx2*, *Bambi*, *Gata3*, and *Dkk1* were increased and abnormally expressed in pharyngeal NCCs (cluster C4) in *Tbx1* null embryos (Fig. 9f, g and Supplementary Data 8). This suggests that an abnormal upregulation of BMP signaling occurred in the absence of *Tbx1*, consistent with data at E9.5 in Fig. 8. GO analysis of the upregulated genes in pharyngeal NCCs (cluster C4) confirms the

**Fig. 5 | *Tbx2* and *Tbx3* are required together for aortic arch artery and arterial branching from the aortic arch. a–c** Intracardiac ink injection of control (**a**) (*n* = 3) and *Wnt1-Cre;Tbx2^f/f^;Tbx3^f/f^* conditional null mutant embryos at E15.5 (**b**, **c**) (*n* = 3). Panel **b** shows a double *Tbx2/Tbx3* cKO embryo with an aberrant retro-esophageal right subclavian artery (ARSA). **d–l** Hematotoxin and Eosin staining on traverse sections of control (**d**, **f**) (*n* = 18) and *Tbx2/Tbx3* cKO mutant embryos (**g–l**) (*n* = 9) at E15.5. Scale bars: 300 μm. Panels **g** and **h** show a *Tbx2/3* cKO embryo with ARSA and panels **j** and **k** show a *Tbx2/3* cKO embryo with a normal right subclavian artery. Panel **i** shows a *Tbx2/3* cKO embryo with aortic arch hypoplasia and panel **l** shows a *Tbx2/3* cKO embryo with a normal aortic arch as compared with control embryos (**f**). **m** Table of the cardiovascular defects in *Tbx2/Tbx3* conditional mutants. ARSA was identified in 5/13 *Tbx2/3* cKO embryos: one after intracardiac ink injection,

three after H&E histology, and one by anatomic analysis after micro-dissection. Aortic arch hypoplasia was found in 2/8 *Tbx2/3* cKO embryos after H&E histology. **n–q** Immunofluorescence on coronal sections of controls at E11.5 (**n**) (*n* = 8) and *Tbx2/3* cKO embryos (**o–q**) (*n* = 8), at the level of the pharyngeal arch arteries for GFP and ACTA2 expression. Nuclei are stained with DAPI. Scale bars: 100 μm. The bottom panels are a high magnification of the dashed regions in **n–q**. Note the strong reduction of ACTA2 expression, ranging from almost total absence to low-level expression, in the right and left fourth PAAs in *Tbx2/3* conditional mutant embryos compared to control embryos (white arrowheads). RSA right subclavian artery, RCC right common carotid, LCC left common carotid, BA brachiocephalic artery, Ao aorta, AoA aortic arch, LSA left subclavian artery, PT pulmonary trunk, E esophagus, PAA pharyngeal arch artery.

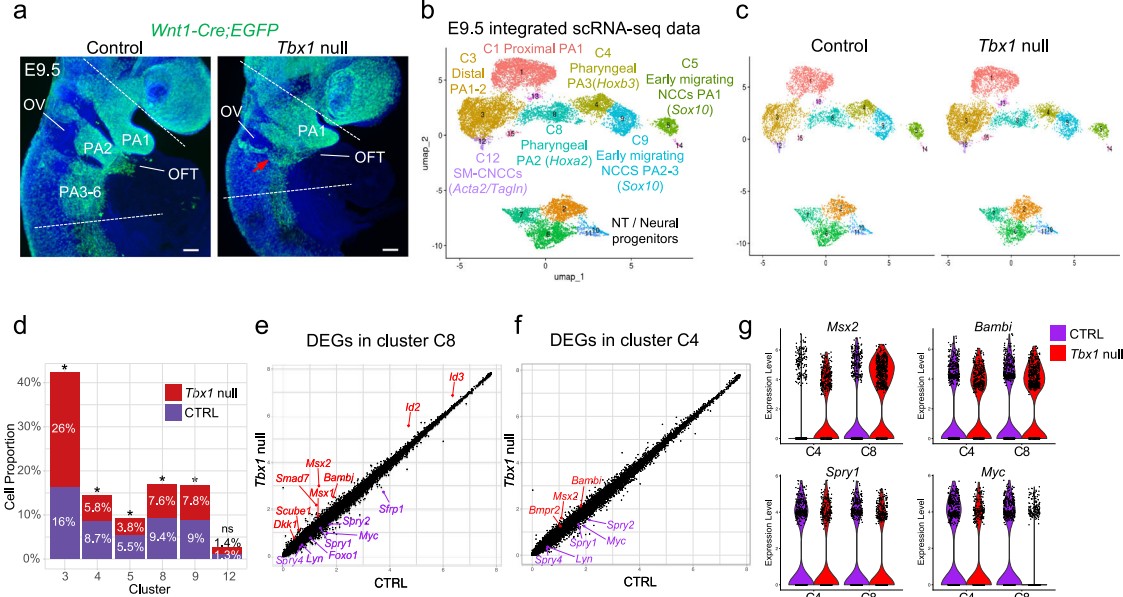

**Fig. 6 | Upregulation of the BMP pathway and downregulation of the MAPK pathway by inactivation of *Tbx1* at E9.5. a** *Wnt1-Cre;ROSA-EGFP* lineage tracing (green) shows mis-localization and a reduced number of NCCs within the pharyngeal region of *Tbx1* null embryos (red arrow) (*n* = 10). DAPI is in blue. NCCs from the region between the two dashed lines were used for scRNA-seq. **b** RISC UMAP plot of integrated scRNA-seq data from NCCs of control and *Tbx1* null embryos. **c** UMAP plots colored by clusters from control (left) and *Tbx1* null embryos (right). **d** Stack bar graph shows proportions of NCCs in indicated clusters divided by the total number of cells in control or *Tbx1* null embryos. A two-sided proportion *z*-test was used to evaluate cell proportion differences between control and *Tbx1* null

embryos (cluster C3: *p* value = 1.14e-56; C4: *p* value = 9.24e-15; C5: *p* value = 1.46e-8; C8: *p* value = 1.23e-5; C9: *p* value = 4.32e-3; C12: *p* value: 0.66) (*p* value <0.05; ns not significantly different). Additional information on statistical analysis is provided in the Source Data files. **e, f** Scatter plots show differential gene expression in cluster C8 (**e**) and in cluster C4 (**f**) from control and *Tbx1* null embryos. Note the upregulation of genes involved in the BMP pathway (red) and the downregulation of genes in the MAPK pathway (purple). **g** Violin plots showing differential expression in control (purple) versus *Tbx1* null (red) for *Msx2*, *Bambi*, *Spry1*, and *Myc* in clusters C4 and C8. PA pharyngeal arch, OV otic vesicle, OFT outflow tract. Scale bars: 100 μm.

upregulation of BMP signaling in *Tbx1* null embryos (Fig. 9h and Supplementary Data 9). By GO analysis, we found downregulation in the expression of genes involved in embryonic organ development and mesenchyme development in pharyngeal NCCs that contain CP-NCCs (Fig. 9i and Supplementary Data 10), suggesting dysregulation of NCC development. We also noted that there was an upregulation of genes that inhibit the cell cycle progression of OFT-CNCCs (Fig. 9j and Supplementary Data 11).

By immunostaining, we confirmed the overall reduction in the number of CNCCs within the OFT and NCCs in the pharyngeal region of *Tbx1* null embryos (Fig. 9k, l). We also confirmed a reduced number of ISL1 + NCCs in the dorsal aortic sac wall mesenchyme and distal OFT, as well as the absence of the aortic sac protrusion in *Tbx1* null embryos (Fig. 9k). Supporting the scRNA-seq data, immunostaining experiments indicated that TBX2 (and likely TBX3) expression was maintained in NCCs located in the lateral part of the pharyngeal apparatus (Fig. 9l). We also noticed that there was normal differentiation of the CNCCs within the OFT of *Tbx1* null embryos despite that there are

fewer cells (Fig. 9l). Together this suggests a reduction but not a complete failure of cardiac fate progression between pharyngeal NCCs and OFT-CNCCs states in the absence of *Tbx1*.

## Discussion

In this report, we identified the cell type signatures and cell fate dynamics of CNCCs. We focused on pharyngeal NCCs in the mouse at developmental stages when *Tbx1*, the gene for 22q11.2DS, is highly expressed and functions. We determined the mechanisms by which *Tbx1* non-autonomously regulates CNCC maturation at a single-cell level and found that altered BMP and FGF-MAPK signaling, together with other signaling, may contribute to cardiovascular malformations when *Tbx1* is inactivated. We also uncovered genes and ligand–receptor pairs with respect to cell-cell communication from mesoderm to NCCs that add to our understanding of CNCC fate progression.

NCCs are multipotent and differentiate into many cell types, including smooth muscle cells. Through examination of transitional

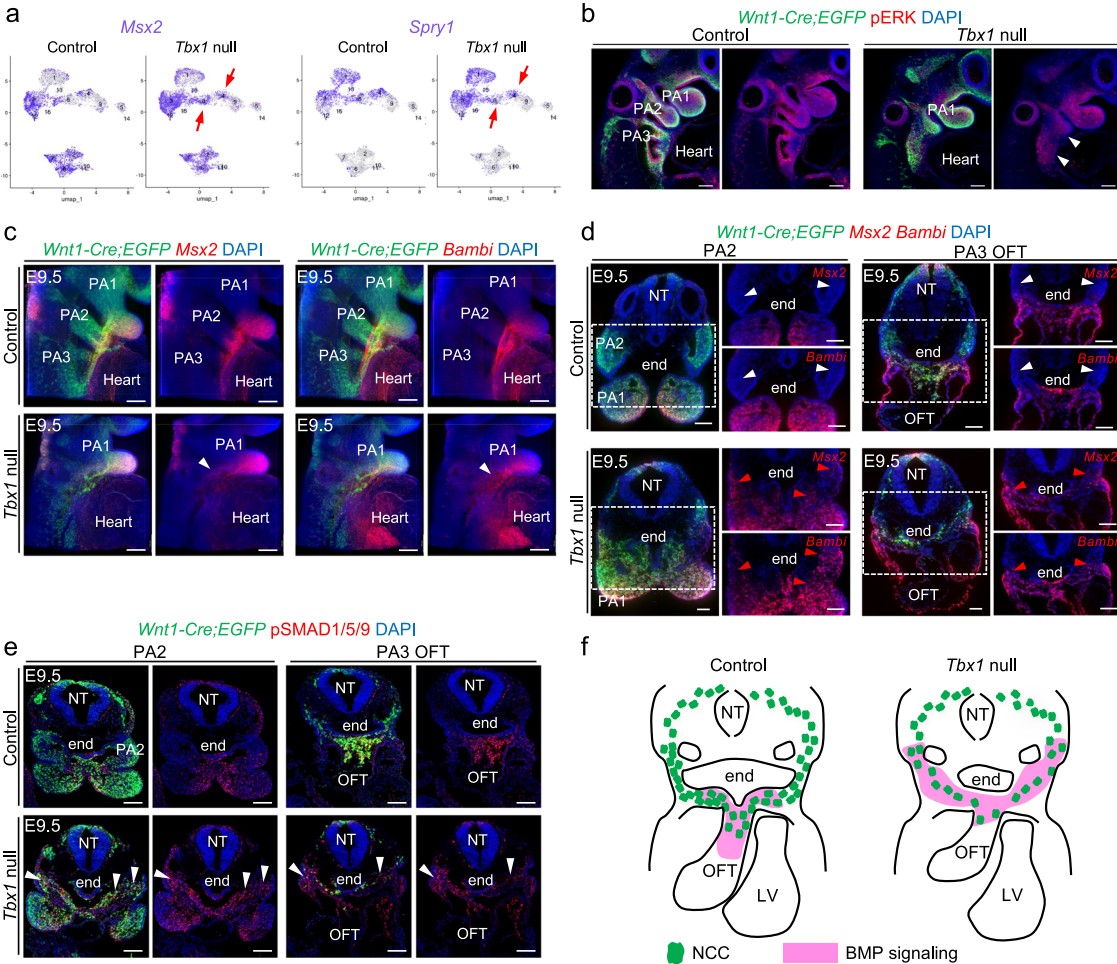

**Fig. 7 | BMP pathway is downregulated and the MAPK pathway is upregulated in pharyngeal NCCs of *Tbx1* null embryos at E9.5. a** UMAP plots show the expression level (purple) of *Msx2* and *Spry1* in NCCs split by control and *Tbx1* null embryos. **b** Wholemount immunostaining through *Wnt1-Cre;ROSA-EGFP* control (*n* = 7) and *Tbx1* null embryos (*n* = 5) for GFP (green) and phospho-ERK (pERK) (red). Digitally generated sections are shown in **b**. DAPI is in blue. Arrowheads show reduced expression of pERK in NCCs in the pharyngeal region of *Tbx1* null embryo. **c** *Msx2*, *Bambi*, and *Egfp* wholemount RNAscope in situ hybridization with DAPI of control and *Tbx1* null embryos. (*Msx2* in control: *n* = 7, *Msx2* in *Tbx1* null embryos: *n* = 3, *Bambi* in control and *Tbx1* null embryo: *n* = 3) The arrowheads indicate ectopic expression of *Msx2* and *Bambi* in *Tbx1* null embryos. **d** *Egfp*, *Msx2*, and

*Bambi* RNAscope assay on transverse sections of control and *Tbx1* null embryos (*n* = 3) showing ectopic and dorsal expression of *Msx2* and *Bambi* in migrating NCCs (red arrowheads). *Msx2* and *Bambi* expression is absent in the proximal part of the PA2 and PA3 of control embryos (white arrowheads). **e** Fluorescent immunostaining for GFP and P-SMAD1/5/9 on transverse sections of control (*n* = 5) and *Tbx1* null (*n* = 4) embryos shows increase in P-SMAD1/5/9 in NCCs of PA2-3 (arrowheads). **f** Schematic representation of control (left) and *Tbx1* null (right) sections at E9.5 showing reduced CNCCs contributing to the shortened OFT and ectopic posterior BMP signaling. PA pharyngeal arch, OFT outflow tract, NT neural tube, end endoderm, LV left ventricle. Scale bars: 100 μm.

dynamics along with embryonic localization by in situ analysis, we uncovered three main transition states from pharyngeal to smooth muscle expressing CNCC derivatives, termed CP-NCCs, OFT-CNCCs, and SM-CNCCs as shown in the model in Fig. 10a.

The CP-NCCs express *Hox* and *Dlx* genes, as well as those implicated in cardiac development, including *Tbx2* and *Tbx3*, as well as other genes (Fig. 10a). Our data also suggests that some *Tbx2/3+* cells differentiate directly into *Acta2+* smooth muscle cells in the PAAs (PAA-CNCCs; Fig. 10a). A role of *Tbx2* and *Tbx3* in CNCCs in PAA development have not been previously described, and we found that inactivation results in the partially penetrant abnormal aortic arch and arterial branching, indicating besides function as markers of CP-NCCs, they have a role in smooth muscle differentiation. Partially penetrant hypoplasia of the aortic arch and arterial branching defects of *Wnt1-Cre;Tbx2ᶠ/ᶠ;Tbx3ᶠ/ᶠ* embryos can result from the ability of the PAAs to recover during development[13,61–65]. Another reason for partial penetrance in mutant embryos could be because mice with a mixed genetic background were used in this study.

The OFT-CNCCs express cardiac transcription factors such as *Hand1/2*, *Msx1/2*, *Foxf1*, and *Gata3*, as well as *Isl1/2* (Fig. 10a). These cells are required for OFT development, as determined by conditional inactivation studies in NCCs of *Hand2*[66], *Msx1*, and *Msx2*[67]. Expression of *Isl1* in CNCCs contributing to the OFT is consistent with a dual lineage tracing study[68]. We also identified genes not previously connected to CNCCs that include *Dkk1*, *Foxf1*, *Rgs5*, *Isl2*, and *Gata3*, as well as newly recognized genes for cardiovascular development.

SM-CNCCs are NCCs within the OFT that expresses *Acta2 and Tagln* smooth muscle genes together with *Gata4/5/6*, *Cnn1*, *Tbx20*, and *Smad6/7*. Supporting their requirement, conditional deletion of *Gata6* in NCCs results in septation defects of the OFT[69]. Abnormal activation of *Smad7* in the NCCs derived from *Wnt1-Cre* genetic lineage leads to pharyngeal arch and cardiac OFT defects[70]. We additionally found genes not yet connected to these cells, including *Meox1*, *Bambi*, and *Smad6*. We found that the progression of CNCC fate toward smooth muscle of the OFT is associated with progressive downregulation of genes involved in pharyngeal embryonic development and

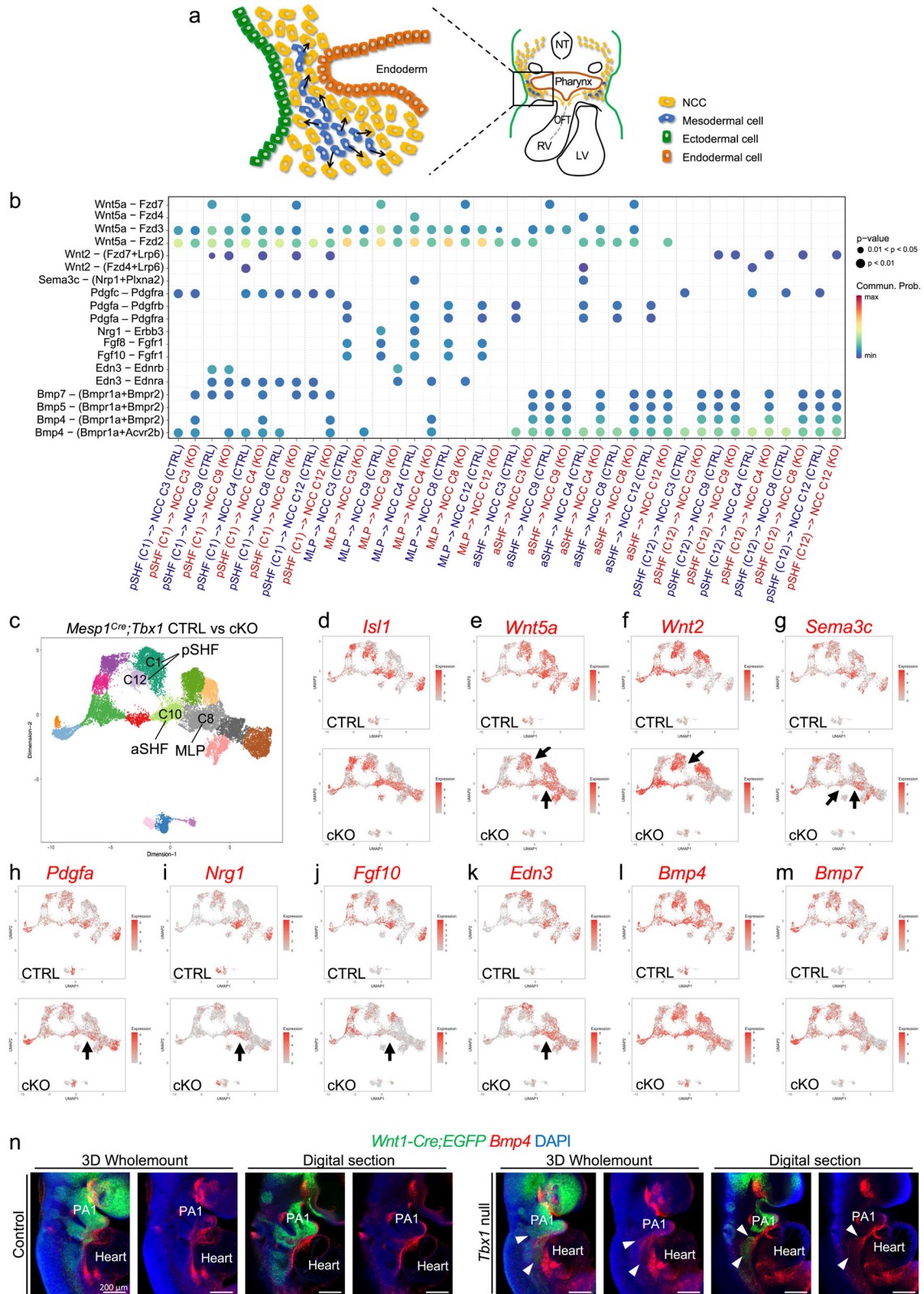

progressive increase in cardiac specification genes, consistent with maturation of the unipotent cellular state to smooth muscle cells (Fig. 10a).

Development of NCCs in the pharyngeal apparatus is regulated by cell-cell signaling, in part from the pharyngeal mesoderm, as uncovered by studies of *Tbx1* mutant embryos[45,71]. In global null or *Mesp1^Cre* mediated *Tbx1* conditional null embryos, there is altered deployment

of CNCCs and reduced contribution to the OFT, leading to a persistent truncus arteriosus[45,71]. Our results suggest that NCCs are produced normally in the neural tube in the absence of *Tbx1*, but they fail to migrate to the caudal pharyngeal arches and OFT, as well as to fail to progress to more mature states, as shown in our model in Fig. 10b. We suggest that failure to migrate and progress in development is due in part to disrupted signaling from adjacent mesodermal cells to the

**Fig. 8 | Cell-cell communication from the mesoderm to NCCs is altered in the absence of *Tbx1* at E9.5. a** Schematic representation of a transverse section showing signaling (arrows) from pharyngeal mesoderm cells (blue) to NCCs (yellow) in the caudal pharyngeal apparatus. **b** Bubble plot shows representative cell-cell signaling from *Mesp1^Cre* derived mesodermal cells to NCCs that were significantly altered (Wilcox rank two-sided test without multiple test correction, *p* value indicated by the size of dot and color; right) in *Tbx1* mutant embryos. Each dot represents a ligand–receptor pair interaction (Y-axis) between a specific cluster in the mesoderm cells and NCCs (X-axis). Clusters include anterior and posterior second heart fields (aSHF, pSHF) and the multilineage progenitors (MLP). **c** UMAP from integration of two replicates of *Mesp1^Cre;ROSA-EGFP* (CTRL) and *Mesp1^Cre;Tbx1*

*^f/f;EGFP* (*Tbx1* cKO) datasets. **d** UMAP plots showing *Isl1* expression in control and *Tbx1* cKO cells. **e–m** UMAP plots showing the expression of ligand genes including *Wnt5a* (**e**), *Wnt2* (**f**), *Sema3c* (**g**), *Pdgfa* (**h**), *Nrg1* (**i**), *Fgf10* (**j**), *Edn3* (**k**), *Bmp4* (**l**), and *Bmp7* (**m**) in *control* and *Tbx1* cKO embryos. Arrows indicate cell clusters with gene expression changes in *Tbx1* cKO embryos. **n** Wholemount RNAscope in situ hybridization of *Wnt1-Cre;ROSA-EGFP* control (lefts panels) and *Tbx1* null (right panels) embryos at E9.5 with probes for *Egfp* and *Bmp*4. Representative digital sections are shown on the right of 3D reconstruction pictures. Arrowheads show expression of *Bmp4* in cells adjacent to the pharyngeal NCCs in *Tbx1* null embryos. PA pharyngeal arch. Scale bars: 200 μm.

NCCs as shown in Fig. 10b. Here, we identified several receptor-ligand interactions that are disrupted by comparing scRNA-seq data from NCCs and the *Mesp1^Cre* lineages.

Using CellChat software[44], we identified signaling interactions from the mesoderm to NCCs and their disruption in *Tbx1* mutant embryos, at a single-cell level. We identified a reduction of FGF ligand expression (*Fgf8* and *Fgf10*) in the mesoderm with reduced MAPK signaling in NCCs, as shown in the model in Fig. 10b. This confirms previous studies showing that inactivation of *Tbx1* results in reduced expression of *Fgf8* and *Fgf10* ligands in the mesoderm[55,72]. It is known that loss of FGF ligands indirectly results in NCC mediated cardiovascular anomalies[42,59,60]. This function is affected significantly by the MAPK signaling pathway that is critical for NCC development and migration[42]. Further, FGF signaling is required downstream of *Tbx1*, both genetically and mechanistically[52,54]. Reduced phospho-ERK1/2 has previously been reported in pharyngeal NCCs of *Tbx1* hypomorphic embryos[73]. Finally, our bioinformatic analysis shows a reduction of downstream effector genes in the MAPK pathway (*Myc*, *Spry1*, and *Spry2*). When taken together, our results support an essential role of mesodermal TBX1-FGF signaling to NCCs that is required for normal heart development (Fig. 10b).

We also found an increase and premature activation of BMP signaling in pharyngeal NCCs of *Tbx1* null embryos along with the expansion of p-SMAD1/5/9 and downstream effector genes, *Msx1/2* and others (Fig. 10b). Inactivation of a BMP antagonist gene, *Dullard* in NCCs resulted in similar cardiac OFT defects as in *Tbx1* null mutant embryos[41] implicating a mechanistic connection between an increase in BMP signaling and OFT defects. The CellChat analysis showed that there is an increase of *Bmp4* and *Bmp7* expression in the adjacent mesoderm relative to the expression of specific *Bmp* receptor genes. Although our RNAscope analysis of *Bmp4* and *Bmp7* did not show an obvious increase in the expression of these two ligands, they continue to be strongly expressed in the mesoderm of *Tbx1* null mutant embryos suggesting that the source of BMP ligands is in adjacent cells that interact with NCCs. It is known that BMP4 can activate BMP signaling through p-SMAD1/5/9[74,75] and this is consistent with our findings (Fig. 10b). In the future, utilization of organoids and cell culture systems will help dissect the mechanism of how BMP interacts with NCCs.

Based upon this analysis, we propose that the inactivation of *Tbx1* results in an alteration of paracrine signaling from the mesoderm, including FGF and BMP, separately contributes to CNCC migration and fate progression, as shown in Fig. 10b. However, these are not the only signaling pathways that are affected by inactivation of *Tbx1*. We found an increase in *Wnt2* expression in the mesoderm by CellChat analysis. We previously found that global *Tbx1* inactivation[76] or inactivation in the aSHF using *Mef2c-AHF-Cre*, resulted in increased *Wnt2* expression[57]. Therefore, canonical WNT signaling also functions downstream of *Tbx1*, and this may also affect NCC function.

The CellChat analysis also revealed downregulation of *Nrg1-Erbb3*, implicating altered Neuregulin to ERBB3 signaling in CNCCs in *Tbx1* mutant embryos. Neuregulin is important for the migration of NCCs, acting as a chemoattractant and chemokinetic molecule[77] and it is

involved in heart development. It has been shown recently that *Nrg1* is a direct transcriptional target gene of TBX1 in the multilineage progenitors (MLPs) in the mesoderm[46]. Therefore, alteration of Neuregulin signaling in *Tbx1* mutant embryos could, in part, explain the failed cardiac contribution of the NCCs. There was other signaling affected, some of which has not been investigated with respect to *Tbx1* or mesoderm-NCC signaling, such as identified by alteration in expression of *Edn3* or *Pdgf*a genes, which can be pursued in the future. Together, our analyses indicate that a combination of important signaling from the pharyngeal mesoderm to NCCs are affected in *Tbx1* mutant embryos and could contribute to the failure of fate progression of CNCCs.

One of the reasons for the disruption of NCCs in *Tbx1* mutant embryos is failed segmentation of the caudal pharyngeal arches by the endoderm. Alternatively, disruption of NCCs could affect the pharyngeal endoderm resulting in failed segmentation. A previous study by Veitch and colleagues showed that after the ablation of NCCs in the chick embryo, pharyngeal segmentation still took place[78]. This suggests that pharyngeal segmentation defects in *Tbx1* null embryos is not due to a reduced number of defective NCCs. There are many reasons for failed segmentation of the caudal pharyngeal apparatus due to *Tbx1* inactivation. The tissue-specific functions of *Tbx1* in this process has been evaluated. Loss of *Tbx1* in the pharyngeal endoderm autonomously affects segmentation of the caudal pharyngeal apparatus leading to reduced contribution of CNCCs[79]. The pharyngeal endoderm is also an important source of signaling during development, including FGF ligands[59], that could potentially affect NCCs development. It was found that *Tbx1* expression in the mesoderm is necessary, non-autonomously, for pharyngeal segmentation[45], suggesting that defects in signaling from the adjacent mesoderm that expresses *Tbx1*, cause failed segmentation. Loss of *Tbx1* in the mesoderm also leads to failed CNCC migration and fate progression. The *Tbx1* expression domain in the pharyngeal ectoderm also serves as a signaling center for non-autonomous signaling[64,80]. It will be interesting to evaluate how the exchange of signaling between the pharyngeal epithelia and NCCs are affected in the absence of *Tbx1*.

In conclusion, in this report, we identified the transcriptional signature that defines the CNCCs and identified the gene expression dynamics that regulate CNCC fate progression into the smooth muscle of the OFT and PAAs. In addition, we highlight the direct alteration of FGF, BMP, and other signaling from the mesoderm to NCCs, most likely leading to failed OFT septation in the absence of *Tbx1* at a single-cell level. Together our results allow a better understanding of the normal development of CNCCs and provide insights into the origin of congenital heart defects associated with defective NCCs and 22q11.2DS.

## Methods
### Mouse lines
Animal experiments were carried out in agreement with the National Institutes of Health and the Institute for Animal Studies, Albert Einstein College of Medicine (https://www.einsteinmed.org/administration/animal-studies/) regulations. The IACUC protocol number is #00001034. The following transgenic mouse lines were used: *Wnt1-Cre*[17], *ROSA26R-GFP^f/+*[18], that we refer to as

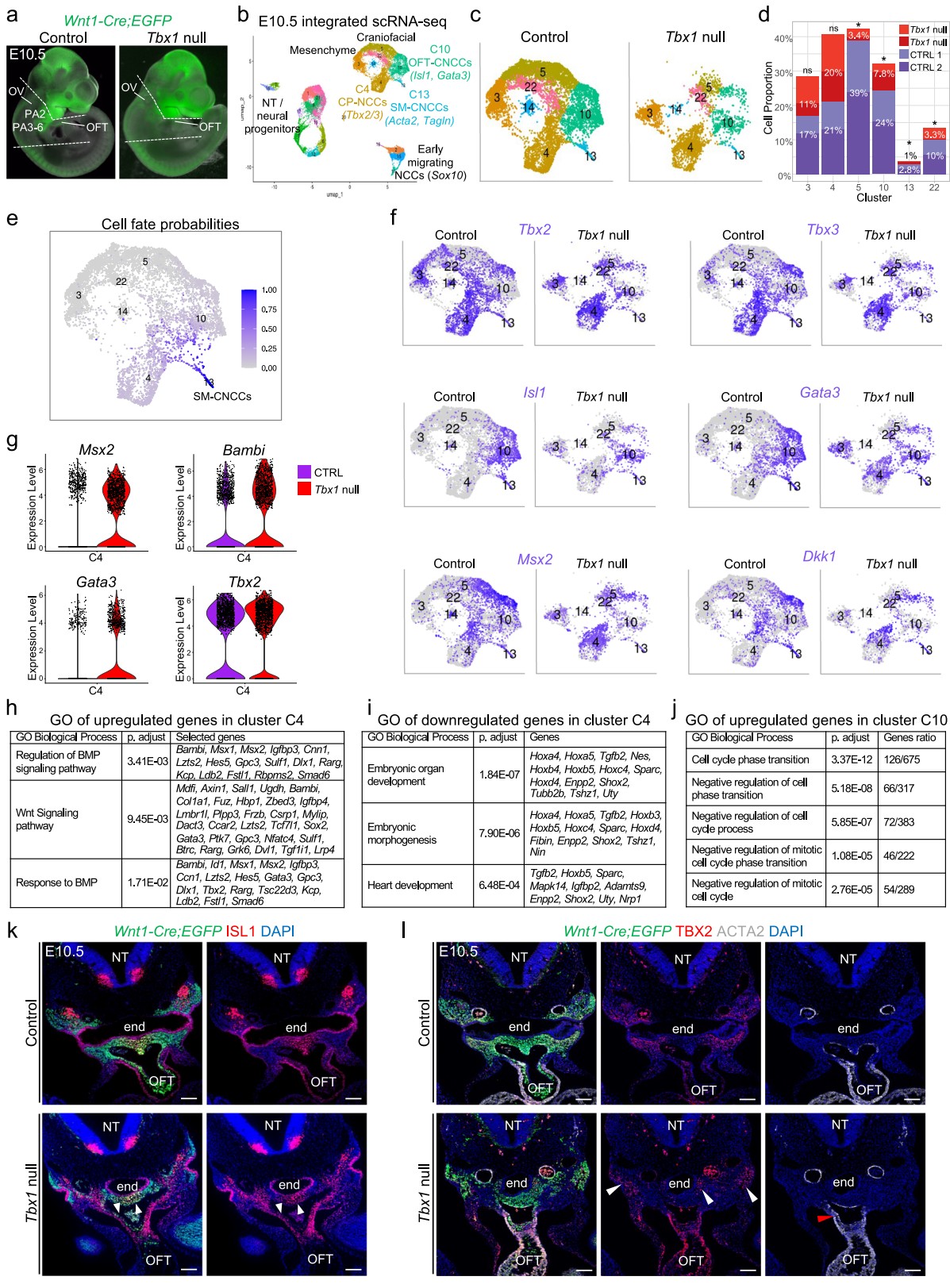

ROSA-EGFP, Tbx1[+/-81], Tbx1[f/+81]. Mice were maintained on a mixed Swiss-Webster genetic background. Tbx2[f/+] and Tbx3[f/+] mutant mouse lines were generated in Dr. Chenleng Cai's laboratory by inserting a two LoxP site into the intron sequences flanking exon 2 of the Tbx2 gene and exons 2–4 of Tbx3 gene, by gene targeting using homologous recombination. After recombination of the genomic sequences between the LoxP sites, the reading frame and T-Box domain of Tbx2 and Tbx3 are both disrupted[82]. Tbx2[f/+] and Tbx3[f/+] mice were maintained on a mixed Swiss-Webster and C57BL/6 genetic background. Embryos were not evaluated for sex, because we studied developmental processes prior to sexual development. Mice and embryos resulting from the different crosses were genotyped from DNA extracted from the tips of toes or yolk sacs. The individual PCR primers and assays used in the

**Fig. 9 | Failure of CNCC cell fate progression by loss of *Tbx1* at E10.5. a** The NCCs between the dotted lines and heart in control and *Tbx1* null embryos were used for scRNA-seq. **b** RISC UMAP plot of integrated scRNA-seq data from NCCs of two replicates of control and *Tbx1* null embryos. **c** UMAP plots of pharyngeal and heart clusters from control (left) and *Tbx1* null (right) embryos. **d** Stack bar graph shows proportions of NCCs in selected clusters divided by the total cell number in control or *Tbx1* null replicates. A two-sided proportion *z*-test was used to evaluate cell proportion differences between control and *Tbx1* null embryos for the two replicates separately if the two *Tbx1* null show consistent increase or decrease compared to control. (**p* value <0.05; ns not significantly different, see Source Data file for *p* value details). **e** UMAP plot colored by cell fate probabilities. **f** UMAP plots showing expression level of *Tbx2*, *Tbx3*, *Isl1*, *Gata3*, *Msx2*, and *Dkk1* in NCCs from control or *Tbx1* null embryos in pharyngeal NCC clusters. *Msx2*, *Gata3*, and *Dkk1*

show increased expression in cluster C4. **g** Violin plots showing differential expression level of *Msx2*, *Bambi*, *Gata3*, and *Tbx2* in cluster C4 in control (purple) and *Tbx1* null (red) embryos. **h–j** GO analysis for upregulated genes in C4 (**h**), downregulated genes in C4 (**i**), and upregulated genes in C10 (**j**). Hypergeometric test was used, adjusted *p* value by Bonferroni–Hochberg correction. **k** Fluorescent immunostaining for GFP (green) and ISL1 (red) with DAPI (blue) on transverse sections of control (*n* = 4) and *Tbx1* null (*n* = 3) embryos. There is reduced ISL1-positive NCCs (arrowheads). **l** Fluorescent immunostaining for GFP (green), TBX2 (red), and ACTA2 (gray) on transverse sections of control (*n* = 4) and *Tbx1* null (*n* = 3) embryos, DAPI is in blue. Expression of TBX2 in NCCs is lateral to the pharyngeal endoderm (end; white arrowheads). There are fewer CNCCs expressing ACTA2 within the OFT of *Tbx1* null embryos (red arrowhead). NT neural tube, end endoderm, OFT outflow tract. Scale bars: 100 µm.

experiments are described in the reports on individual mouse alleles. Embryos were collected at different embryonic days dated from the day of the vaginal plug (E0.5). For each experiment, a representative result was presented from at least three analyzed embryos.

## Immunofluorescence staining on paraffin sections

Embryos were collected in cold 1x PBS (Phosphate Buffered Saline) and fixed in 4% PFA (Paraformaldehyde) for 1 h at 4 °C under constant agitation. Embryos were progressively dehydrated in ethanol, then xylene, and embedded in paraffin (Paraplast X-tra, Sigma P3808). The embryos were sectioned to the 10-µm thickness and sections were deparaffinized in xylene and progressively rehydrated in an ethanol series. Sections were incubated and boiled in antigen unmasking solution (Vector laboratories, H-3300) for 15 min. After cooling at room temperature, sections were washed in PBS containing 0.05% Tween (PBST) and blocked for 1 h in TNB buffer (0.1 M Tris-HCl, pH 7.5, 0.15 M NaCl, 0.5% Blocking reagent [PerkinElmer FP1020]) at room temperature. Then sections were incubated with primary antibodies diluted in TNB overnight at 4 °C. Sections were washed in PBST and incubated with secondary antibodies diluted in TNB for 1 h at room temperature. After washes in PBST, nuclei were stained with DAPI (1/1,000; Thermo Scientific, 62248) and slides were mounted in Fluoromount (Southern Biotech). Embryonic sections were imaged using a Zeiss Axio Imager M2 microscope with ApoTome.2 module. Images were analyzed using Zen software from Zeiss, ImageJ, and Adobe Photoshop.

## Immunofluorescence on wholemount embryos

Embryos were collected in 1x PBS and fixed in 4% PFA overnight at 4 °C under constant agitation. Then embryos were washed in 1x PBS and progressively dehydrated in a methanol bath series (25, 50, and 100%) and stored at −20 °C. After progressive rehydration in 1x PBS, embryos were incubated in 1x PBS containing 0.1% Triton (PBS-Triton) overnight at 4 °C. Embryos were then incubated in blocking solution (1x PBS containing 10% Donkey serum (Sigma, D9663)) overnight at 4 °C under continuous agitation. Then embryos were incubated with primary antibodies diluted in blocking solution for 2 days at 4 °C under constant agitation. After 5 watches (1 h each) in 0.1% PBS-Triton at 4 °C, embryos were incubated with secondary antibodies, and DAPI diluted in a blocking solution for 2 days at 4 °C. Embryos were washed in 0.1% PBS-Triton five times, then progressively dehydrated in 100% methanol and cleared in benzyl alcohol-benzyl benzoate (2:1 volume per volume ratio) (BABB) solution. Embryos were imaged as a Z-stack using a Nikon CSU-W1 spinning disk confocal microscope. Images were analyzed using ImarisViewer 9.8.0 software.

## Antibodies used for immunostaining experiments

The following primary antibodies were used: goat anti-GFP (1/200, Abcam ab6673), mouse anti-alpha smooth muscle actin ACTA2) (1/

200, Abcam ab7817), rabbit anti-αSMA (ACTA2; 1/200, Abcam ab5694), mouse anti-Isl1/2 (1/100, DSHB 40.2D6 and DSHB 39.4D5), mouse anti-TBX2 (1/100, Santa Cruz sc514291), rabbit anti-TAGLN (1/200, Abcam ab14106), rabbit anti-p-SMAD1/5/9 (1/100, Cell Signaling D5B10 #13820), rabbit anti-TBX1 (1/100, Lifescience LS-C31179), rabbit anti-pERK (1/100, Cell Signaling p44/42 MAPK, 9101), and MF20 (1/100, Sarcomeric Myosin heavy chain antibody, mouse, DSHB). The following secondary antibodies were used from Invitrogen (Thermo Fisher Scientific) raised in donkey at 1/200: anti-goat IgG Alexa Fluor 488 (A11055), anti-mouse IgG Alexa Fluor 555 (A31570), anti-mouse IgG Alexa Fluor 647 (A3157), anti-rabbit IgG Alex Fluor 555 (A31572), anti-rabbit IgG Alex Fluor 568 (A10042) and anti-rabbit IgG Alex Fluor 647 (A31573).

## RNAscope

**RNAscope on wholemount embryos.** Embryos were collected and dissected in 1x PBS at 4 °C and fixed in 4% PFA overnight. Then embryos were dehydrated in progressive methanol washes and stored in 100% methanol at −20 °C. Wholemount RNAscope was performed using RNAscope Multiplex Fluorescent Detection Reagents v2 kit (Advanced Cell Diagnostics, ref 323110). Embryos were progressively rehydrated in 1x PBS containing 0.01% Tween (PBST) and were permeabilized using Protease III (Advanced Cell Diagnostics, ref 323110) for 20 minutes at room temperature, followed by washes in PBST. Embryos were then incubated with 100 µl of pre-warmed mixed C1, C2, and C3 (ratio 50:1:1, respectively) RNAscope probes at 40 °C, overnight. After three washes in 0.2x SSC (Saline Sodium Citrate), 0.01% Tween, embryos were fixed in 4% PFA for 10 min at room temperature. Embryos were then incubated in Amp1 for 30 min at 40 °C, Amp2 for 30 min at 40 °C, and Amp3 for 15 min at 40 °C, with washes in 0.2x SSC, 0.01%Tween between each step. Tyramide signal amplification (TSA) solutions were prepared as follows: 1/500 for TSA-Fluorescein (Akoya Biosciences, NEL741001KT), 1/2000 for TSA-CY3 (Akoya Biosciences, NEL744001KT), and 1/1,000 for TSA-CY5 (Akoya Biosciences, NEL745001KT). To reveal C1 probes, embryos were incubated in HRP1-C1 for 15 min at 40 °C, then washed and incubated with the chosen TSA solution, for 30 min at 40 °C. The amplification reaction was blocked using HRP-Blocker for 15 min at 40 °C. C2 and C3 probes were revealed following the previous steps and using HRP-C2 for C2 probes and HRP-C3 for C3 probes. Nuclei were stained overnight with DAPI. Wholemount embryos were imaged as Z-stacks using a Leica SP5 confocal microscope or a Nikon CSU-W1 spinning disk confocal microscope, after being cleared with BABB or not. 3D image reconstruction and analyses were performed using ImageJ and ImarisViewer 9.8.0 software.

**RNAscope on cryosections.** Embryos were collected and dissected in 1x PBS at 4 °C then fixed in PFA 4% overnight and incubated in successive 10, 20, and 30% sucrose (Sigma-Aldrich S8501) solutions and then embedded in OCT (Optimal Cutting Temperature compound). Embryos were stored at −80 °C until they are used. RNAscope was

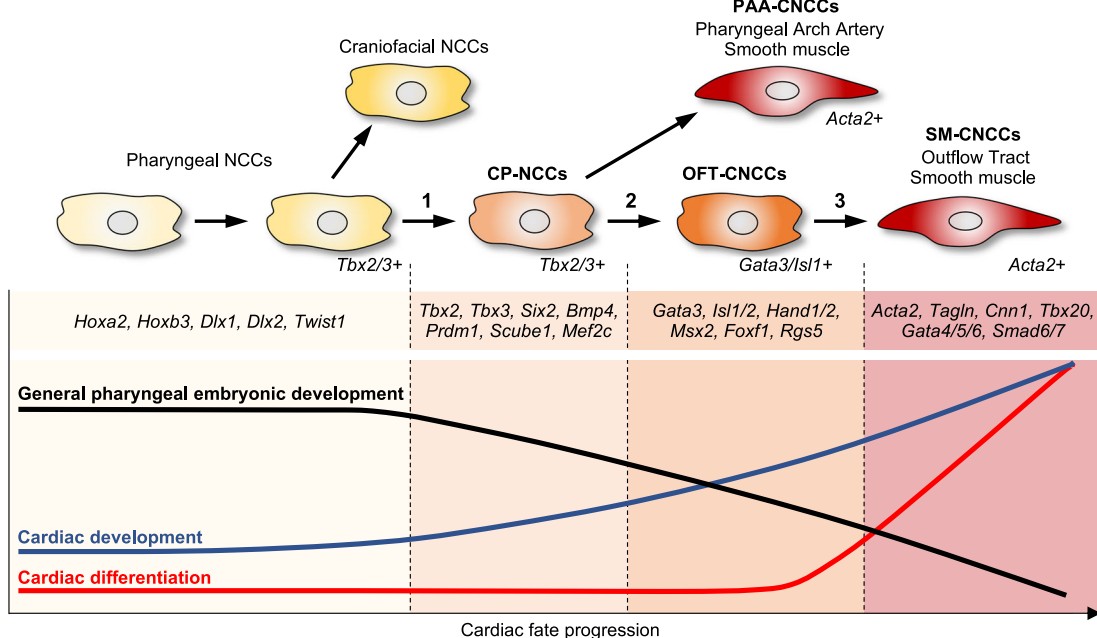

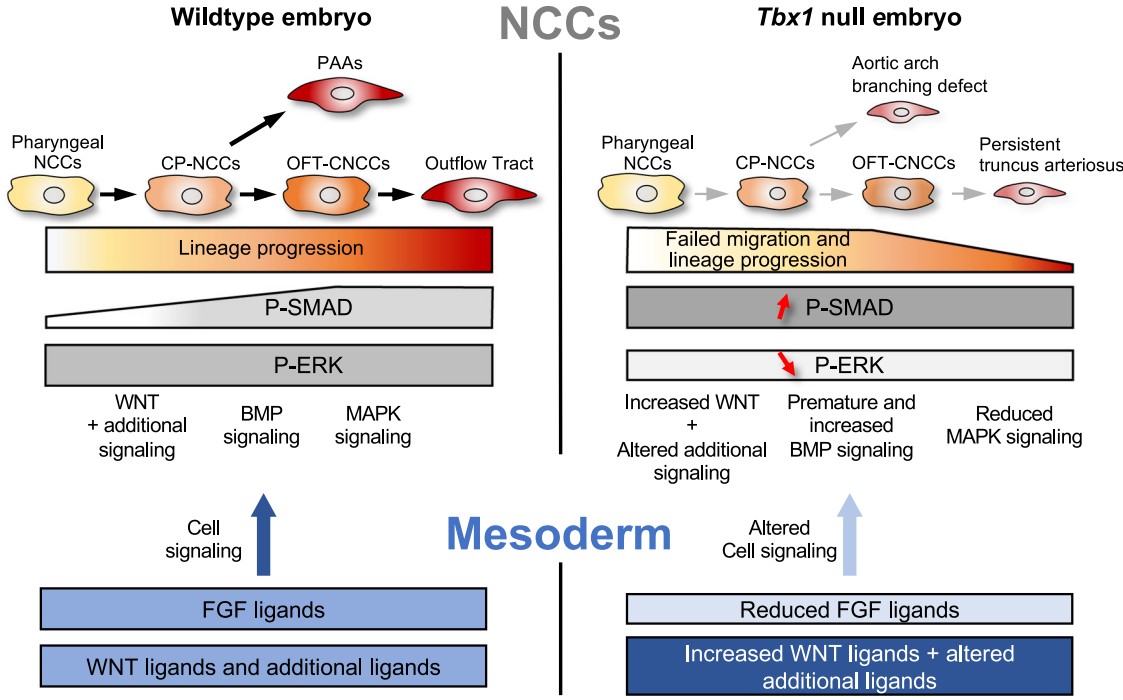

**Fig. 10 | Model of a multistep specification to form CNCCs and the signaling pathways disrupted by inactivation of *Tbx1*. a** Colors indicate cell fate progression of pharyngeal NCCs (light yellow) towards vascular smooth muscle cells (red). Step 1 shows the transition between pharyngeal NCCs expressing *Tbx2/3* to cells expressing markers of CNCCs (CP-CNCCs). Step 2 shows the transition in which some cells directly differentiate to smooth muscle of PAA-CNCCs, while the majority migrate and enter the OFT as OFT-CNCCs and express *Gata3/Isl1* and Step 3 shows the transition to SM-CNCCs expressing *Acta2*. The graph shows the relative change of biological gene ontology terms during the three transitions of CNCCs to smooth muscle cells. **b** Model of non-autonomous signaling between the mesoderm (bottom, blue) and NCCs (top, gray) highlighting FGF, WNT, and additional signaling from the mesoderm to NCCs in normal embryos (left) and how signaling is disrupted in the absence of *Tbx1* (light blue arrows/inhibitory arrows; right). Alteration of signaling in the mesoderm in the absence of *Tbx1* results in abnormal premature activation of the BMP pathway during NCC deployment and migration (light blue inhibitory arrow; red, up arrow), a decrease of MAPK signaling in NCCs (light blue arrow; red down arrow), and alteration of other signaling pathways including upregulation of WNT signaling in developing NCCs. This leads to failure of cardiac fate progression with a reduced number of CP-CNCCs, reduced number of CNCCs, and reduced number of SM-CNCCs (gray arrows) with aortic arch/branching defects and failed OFT septation (persistent truncus arteriosus).

performed on 10-μm sections mounted on SuperFrost Plus slides (Fisher Scientific, 12-550-15) following the RNAscope Multiplex Fluorescent Reagent Kit v2 assay protocol from Advanced Cell Diagnostics. Sections were imaged using a Zeiss Axio Imager M2 microscope with ApoTome.2 module. Acquisitions were processed using Zen software from Zeiss, ImageJ, and Adobe Photoshop.

Probes used for RNAscope: *Egfp* (400281, 400281-C3), *Tbx2* (448991), *Hoxb3* (515851), *Dlx2* (555951), *Bambi* (523071), *Hoxa2* (451261), *Gata3* (4033321-C2), *Isl1* (451931, 451931-C2), *Tbx3* (832891-C2), *Sox10* (435931-C2), *Msx2* (421851-C2), *Meox1* (530641-C2), *Dlx5* (478151-C3), *Twist1* (414701), *Rgs5* (430181-C3), *Cnn1* (483791), *Bmp4* (401301), and *Bmp7* (407901).

### Histology and staining with Hematoxylin & Eosin

Fetuses were collected and dissected in 1x PBS and fixed overnight in 4% PFA. They were progressively dehydrated in ethanol and incubated in xylene prior to embedding in paraffin. Tissue sections of 12-μm thickness were deparaffinized in xylene and progressively rehydrated in ethanol washes and incubated for 10 min in Hematoxylin (Sigma-Aldrich, HHS16) then rinsed in water and dehydrated in 70% ethanol. Sections were then incubated in alcoholic Eosin (70%) (Sigma-Aldrich, HT110116) solution and progressively dehydrated in ethanol and xylene washes prior to mounting in Permount mounting medium (Fisher Chemical SP15100). Sections were imaged using a Zeiss Axioskop 2 plus microscope and analyzed using Zen software and Adobe Photoshop software.

### Intracardiac India ink injection

Fetuses at E15.5 were dissected in 1x PBS and the chest was carefully opened to avoid damaging the cardiovascular system. A solution containing 50% India ink and 50% 1x PBS was injected into the left ventricle of the heart by blowing gently into an aspirator tube assembly connected to a microcapillary. Immediately after filling the left ventricle and arterial branches, the heart and aortic arch with arterial branches were imaged using a Leica MZ125 stereomicroscope. Images were analyzed using Zen software and Adobe Photoshop software.

### scRNA-seq data generation

Embryos were collected and microdissected in 1x PBS at 4 °C. Dissected tissues of interest were maintained in DMEM (Dulbecco's Modified Eagle Medium, GIBCO 11885084) at 4 °C. For E8.5 embryos, the rostral half of the embryos, including the heart, was collected. The head of the embryo was conserved at E8.5 because of the small size of the embryos that most likely contain CNCC progenitors in the anterior part of the pharyngeal region. At E9.5, the pharyngeal region from PA1 onwards, plus the heart, were collected. The head of the embryos containing NCCs not relevant to the cardiovascular system were removed at E9.5 to not dilute out the pharyngeal NCCs, including CNCCs. At E10.5, the pharyngeal region plus the heart were collected. The head and PA1 were removed at E10.5, as shown in Fig. 1, because, at this time point, these NCCs are destined to form the craniofacial skeleton, not the cardiovascular system. Then, tissues were incubated in 1 ml of 0.25% Trypsin-EDTA (GIBCO, 25200056) containing 50 U/ml of DNase I (Millipore, 260913-10MU), at room temperature for 7 min. Next, heat-inactivated FBS (fetal bovine serum, ATCC, 30-2021) was added to stop the reaction at a final concentration of 10%, at 4 °C. Dissociated cells were centrifuged for 5 min at $300 \times g$ at 4 °C and the supernatant was removed. Cells were then resuspended in 1x PBS without $Ca^{2+}$ and $Mg^{2+}$ (Coming, 21-031-cv) containing 10% FBS at 4 °C and filtered with a 100-μm cell strainer. A total of 1 μl DAPI (1 mM) (Thermo Fisher Scientific, D3571) was added. Cells were FACS (Fluorescence Activated Cell Sorting) purified using a BD FACSAria II (BD Biosciences) system and BD FACSDiva 8.0.1 software in order to collect EGFP-positive and DAPI negative living cells in 1x PBS containing 10% FBS. The FACS gating strategy is exemplified in Supplementary Fig. 15.

EGFP positive, DAPI negative cells were then centrifuged at 4 °C, 5 min at $300 \times g$ and resuspended in 50 μl of 1x PBS without $Ca^{2+}$ and $Mg^{2+}$ with 10% FBS. Cells were then loaded in a 10x Chromium instrument (10x Genomics) using the Chromium Single Cell 3′ Library & Reagent kit v2 or single index Chromium Next GEM Single Cell 3′ GEM, Library & Gel Bead kit v3 or v3.1. One scRNA-seq experiment was performed for control and *Tbx1* null embryos at E9.5 and two biological replicates were performed for control and *Tbx1* null embryos at E10.5. Each sample that was submitted for scRNA-seq was a pool of two or more embryos of the same genotype (Supplementary Table 1)

### Sequencing

Sequencing of the DNA libraries was performed using an Illumina HiSeq 4000 system (Genewiz Company, South Plainfield, NY, USA) (scRNA-seq at E9.5 and E10.5) or NovaSeq 6000 system (Novogene, Sacramento, CA, USA) (scRNA-seq at E8.5) with paired-end, 150 bp read length.

### scRNA-seq data analysis

CellRanger (v6.0.1, from 10x Genomics) was used to align scRNA-seq reads to the mouse reference genome (assembly and annotation, mm10-2020-A) to generate gene-by-cell count matrices. All the samples passed quality control measures for Cell Ranger version 6.0.1 (Supplementary Table 1).

**Seurat analysis for filtering and clustering.** Individual scRNA-seq sample data were analyzed using Seurat V4.0.5[19], with parameters as recommended by Seurat software.

**Integrated scRNA-seq analysis.** After individual samples were analyzed by Seurat for clustering, the data were integrated by RISC software (v1.6, https://github.com/bioinfoDZ/RISC) using the Reference Principal Component Integration (RPCI) algorithm for removing batch effects and aligning gene expression values between the control and *Tbx1* null samples at E9.5 and E10.5[40]. The integrated data were re-clustered by RISC, using parameters adjusted to match the cell type clusters in the Seurat results. Gene expression differences between control and *Tbx1* null embryos was determined by RISC software for each of the clusters at an adjusted $p$ value <0.05 and log2 (fold change) >0.25 (using a negative binomial model). Differentially expressed genes detected in at least 10% of the cells of the clusters of interest were included and those with lower percentages were excluded from the analysis. Cell compositions were computed from the integrated cell clusters and used for the two-proportion $Z$-test as implemented in the R prop.test() function to evaluate the statistical significance in changes between control and null datasets. At stage E9.5, scRNA-seq was performed with one replicate of control and one replicate of *Tbx1* null, with each containing more than one embryo of each genotype. At stage E10.5, we compared the two replicates of control and *Tbx1* null scRNA-seq datasets separately and only performed this statistical test for the cell clusters if the directions of the changes were consistent in the two replicates.

**Cell trajectory analysis.** CellRank (v1.5.1)[32] was used to infer differentiation trajectory, focusing on determining the probability of cells to adopt the smooth muscle, *Acta2* + cell fate. The analysis used RNA velocity from Velocyto (v0.17.17)[83] and scVelo (v0.2.4)[83], as well as cell-cell similarity, to infer trajectories and cell differentiation potential. The analysis was performed for all cells in either the E9.5 control sample or the two E10.5 control samples and then for the cells in the selected clusters that were predicted to have connections to the smooth muscle *Acta2* + cluster.

**Cell-cell communication analysis.** CellChat (v1.1.3)[44] was used to identify the ligand–receptor interactions between *Mesp1*<sup>*Cre*</sup> mesoderm

lineage and NCC lineages and then compare the change between control and *Tbx1* null mutant data at p < 0.05. Data included ligands in the *Mesp1^Cre* lineage and receptors in the CNCC lineage. *Mesp1^Cre* control and *Tbx1* conditional null datasets were integrated with RISC software for clustering and dimension reduction using cell type annotation as previously described in ref. [46].

## Reporting summary

Further information on research design is available in the Nature Portfolio Reporting Summary linked to this article.

## Data availability

All scRNA-seq datasets generated in this study have been deposited in NIH GEO (Gene Expression Omnibus) under the accession number: GSE210521. All other data supporting the findings of this study are available within the article and its Supplementary Information files or from the corresponding authors upon request. Source data are provided with this paper.

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

## Acknowledgements

We thank members of the Genomics core, Flow Cytometry, and Analytical Imaging facilities at Albert Einstein College of Medicine (NCI Cancer Center Support Grant P30CA013330). We are grateful to Professor Chenleng Cai for *Tbx2*[f/+] and *Tbx3*[f/+] mice. We thank Professor Robert G. Kelly for his insightful comments on the manuscript. This work was supported by grants from the National Institutes of Health [P01HD070454 (B.E.M. and D.Z.), R01HL138470 (B.E.M.), R01HL153920 (B.E.M. and D.Z.), R01HL163667 (B.E.M. and D.Z.)] and by a grant from the Fondation Leducq (Transatlantic Network of Excellence 15CVD01; B.E.M. and C.D.B.). C.D.B. thanks the Fondation Bettencourt-Schueller and the Philippe Foundation for their financial support.

## Author contributions

C.D.B. and B.E.M. designed the study and experiments. C.D.B. performed all wet laboratory experiments. C.D.B., Y.L., A.F., and A.V. performed computational analysis of single-cell RNA-sequencing data. Y.L., A.F., and D.Z. provided bioinformatics expertize and guidance. C.D.B. and B.E.M. wrote the manuscript. B.E.M. secured funding for the project. All authors read, intellectually contributed, edited, and approved the manuscript.

## Competing interests

The authors declare no competing interests.
