## [Peer Review File · Nature Communications]

Single-cell transcriptomics uncovers a non-autonomous Tbx1-dependent genetic program controlling cardiac neural crest cell developmentREVIEWER COMMENTS

Reviewer #1 (Remarks to the Author):

In this manuscript, De Bono and colleagues investigate single-cell transcriptomic profiles of neural crest cells from the pharyngeal apparatus to the heart in normal and Tbx1-null embryos. They demonstrate transitions of NCC differentiation stages characterized by different transcription factors, which are affected non-autonomously by loss of Tbx1. Among affected transcription factors, co-inactivation of Tbx2 and Tbx3 caused aberrant right subclavian artery with incomplete penetrance. Transcriptomic profiles further showed upregulation of BMP signaling and reduced MAPK signaling in CNCCs of Tbx1-null embryos, which lead them to argue that alteration of mesoderm-derived FGF signaling may result in excessive BMP activation responsible for cardiac malformations in 22q11.2 deletion syndrome. Overall, data are generally of high quality and are well presented. However, the data are largely descriptive and correlative in nature and the mechanistic novelty is limited and lacking direct validation especially for the source of responsible BMP ligands and the involvement of altered BMP and MAPK signaling pathways in the Tbx1-null phenotype.

Specific comments:

1. Regarding the single-cell transcriptomic analysis, Chen et al. have published a paper showing partially overlapping data of CNCCs with diverse lineages (EMBO rep. 2021), which should be referred to in the manuscript and discussed together.
2. The authors characterize Acta2-expressing cells as smooth muscle cells. However, Acta2 is also expressed in different cell types such as myofibroblasts, pericytes and muscle progenitors during embryonic development. Indeed, Acta2 expression is found in NCCs migrating into the endocardial cushion at E10.5 (Fig. 3F), whose phenotype is likely immature mesenchyme. The authors should characterize Acta2-expressing cells more definitely with several cell type markers.
3. In the PHATE map of E9.5 CNCCs (Fig. 2H), clusters are allocated to different regions. For example, C5 and C3 likely correspond to PA2 and PA3, respectively. Then, how did the authors construct the heatmap in Fig. 2K along the latent time? Is it derived from a mixture of data from spatially different clusters? If so, how is it validated? There may be a similar concern for the analysis shown in Fig. 3, where the clusters are not spatially allocated but different Hox gene expression patterns indicate the clusters might be regionally distinct cell populations.
4. About 40% of Tbx2/3 cKO embryos have an aberrant right subclavian artery but no other defects. Based on immunostaining shown in Fig. 4J-L, the authors discussed the role of Tbx2/3 in smooth muscle differentiation. Is the failed smooth muscle differentiation of the right fourth PAAs due to defects in differentiation capacity or delayed NCC deployment and/or differentiation? Are the aberrant right subclavian arteries defective for the smooth muscle layer? The pictures also show low Acta2 signals in the left PAAs as compared with the right PAAs. Were no defects observed in the left fourth PAA-derived aortic arch?
5. How about the ductus arteriosus derived from the left sixth PAA?
6. Regarding the analysis on Tbx1 mutants, presented data clearly demonstrate differentially expressed genes between wild-type and Tbx1-null mutant embryos, but mechanistic relationships are speculative. Although several ligand-receptor pairs were extracted as altered signals in Tbx1 mutants, direct evidence is lacking as for their contribution to the Tbx1-null phenotype. Rescues of the Tbx1-null phenotype by conditional genetic inactivation or activation of BMP or MAPK signaling pathways are expected to give a direct support for their contribution. It is also important to identify the source of BMP ligands responsible for the Tbx1-null phenotype.
7. What is the relationship between clusters shown in Fig 1E and Fig 5B? Then, is it possible that altered BMP and MAPK signaling pathways affects transcriptomic profiles of each CNCC cluster?

Reviewer #2 (Remarks to the Author):

This is a detailed and comprehensive paper describing the transcriptome of cardiac neural crest cells that migrate through the pharyngeal region and to the outflow tract during a critical time of cardiovascular development in the mouse. The transcriptome of neural crest cells from Tbx1-null embryos has also been analysed and key gene expression differences identified.

There are, however, some issues that need clarification.

The introduction should briefly describe the formation and remodelling of the PAA, particularly as this information is required to understand the ARSA phenotype described later. Also, describe the role of NCC in outflow tract septation as this is not restricted to smooth muscle formation.

Based on the data acquired from control embryos, the authors tested the role of Tbx2 and Tbx3 in cardiovascular development by conditionally deleting both genes simultaneously from NCC. A proportion of mutant fetuses have ARSA.

Line 331: mentions 'reduced penetrance'. What is this compared to? Or should this be referring to a low penetrance?

The legend for Figure 4 states that there is a "...strong reduction of Acta2 expression in the right 4th PAA...in mutant embryos...". The implication is that it is the right PAA only that is lacking smooth muscle cell investment which causes the ARSA phenotype. Panel K and L focus on the right 4th PAA in mutants, but the left PAA (which is phenotypically unaffected at E15.5) looks to have a similar level of smooth muscle staining which would suggest that the effect (if any) is symmetrical and equal for each 4th PAA.

How many mutant embryos were analysed? Were they stage-matched with the controls? Methods suggest at least 3 were analysed but this needs to be clarified, particularly as only 38.5% of fetuses develop the ARSA phenotype. If all embryos have reduced 4th PAA smooth muscle staining, why do the majority of fetuses have a normal right subclavian artery?

The coronal sections suggest that 4th PAA formation occurs – the vessel is well formed - so can the authors please discuss what they think happens to this vessel? For example, do they suspect that a lack of smooth muscle cell investment causes the remodelling vessel to collapse, resulting in ARSA? Overall, this part of the study needs more discussion. E.g. why is there a low penetrance of 4th PAA defects, why is the right 4th PAA affected and not the left, what happens to the vessel if there are fewer smooth muscle cells, etc.

Although this part of the study is independent of the Tbx1 analysis, this group has previously published that Tbx1-hets on a Swiss-Webster background do not have 4th PAA defects, unlike on other backgrounds such as C57Bl/6. Could the low penetrance of 4th PAA defects in the Tbx2/3 mutants be related to genetic background?

Line 513: Suppl Table 1: why are the E10.5 stages split into two groups (there are two sets of control and two sets of mutant)? Were these samples pooled for analysis?

Suppl Table 1 suggests 20 embryos were used altogether. The manuscript says that 36,721 NCCs were sequenced (line 85). This doesn't match up with the data from the 20 embryos in Suppl Table 1 which says a total of 65,852 NCCs were sequenced.

Line 206-209: Why are there fewer NCC in Tbx1-nulls? Conversely, Figure 5D shows that there more NCC in PA1-2 in Tbx1-nulls compared to controls. Is this related to the failure in pharyngeal segmentation preventing NCC migration, or do fewer NCC cause the segmentation defect? Can the authors speculate why this might be?

Given that Tbx1 is expressed in ectoderm, endoderm and the mesoderm of the pharyngeal arches, signalling from one or more of these tissues must be influencing NCC behaviour. The discussion only briefly mentions the possibility that the endoderm may also be involved in Tbx1 signalling so this could be expanded on to include the ectoderm. Mutant mouse studies have been published to show that Tbx1 genetically interacts with other genes in these tissues to regulate PAA morphogenesis.

Line 199 Authors state 'We found that Tbx1 was not noticeably expressed in NCCs'. However, there are clearly some NCCs positive for Tbx1 at E8.5 in Suppl. Fig 2 and some at E9.5. Although the authors present data that shows deleting Tbx1 in NCC does not lead to cardiovascular defects, these few groups of double positive cells should be labelled, despite their being a small number.

Why is there no supporting RNA scope for Twist1 in Figure 1? Figure 1G-I focuses on the decrease in Sox10 expression from E8.5-E10.5 and the simultaneous increase in Twist1 expression, however, only Sox10 RNA expression is validated with the RNA scope. As this is an important correlation Twist1 RNA should also be validated/visualised in this way. Just as Hox2a and 3b have also been validated in this Figure.

Minor comments

Line 63: "Est1" should be "Ets1"

Line 154: states 'This data also indicates that a small fraction of P-CNCCs may directly differentiate to smooth muscle cells (blue fraction in the pie chart in C2)'. This reads as though the authors are suggesting some cells in cluster 2 differentiate directly to cells in cluster 10, an arrow depicting this should be included in Figure 3K. If the authors are referring to cells in C2 directly differentiating to cells in C4 then this isn't clear and should be reworded in the manuscript.

Line 170: authors state 'addition, our functional enrichment analysis identified genes associated with congenital heart disease such as tetralogy of Fallot, double outlet right ventricle and ventricular septal defects (Supplementary data 6)'. Looking at this data, it was in group 4 only of the segregated NCCs where this was identified, therefore this should be clarified in the manuscript.

Line 269: Authors state 'Two additional ligand genes, Edn3 and Pdgfa were not investigated regarding Tbx1' which reads as if the authors chose not to investigate these and should be written as 'To our knowledge there is no record of Edn3 and Pdgfa having been investigated in relation to Tbx1'.

Line 402: could be written more clearly, using the term recombined.

Line 404: should be C57BL/6

Line 487: Refer to Suppl Table 1 as this contains the details of the number of embryos collected for analysis.

Line 498: More detail required for how the FACS was performed.

Line 499: 'DAPI negative' should be 'DAPI positive'

Line 533: Number of samples – implies that n=1 at E9.5 and n=2 at E10.5 were analysed. This doesn't fit with Suppl Table 1 details.

Line 823: Cluster C5 and C9 not labelled in Figure 5B, but these are shown in 5D.

Line 137 referring to Figure 2G: The authors state 'the first CNCCs from PA3 were entering the OFT (Fig. 2G).' However, Fig 2F does look to show some CNCCs in the OFT coming from PA3. The authors should clarify this statement to say there are a small number of early CNCC in the OFT from PA3, however, the stream of CNCCs from PA3 is fully established in 2G.

Figure 1: legend has no N numbers

Figure 1: Sox10 changes colour from red to blue to red again across Figure 1J-L which is confusing, can this be modified to be kept consistent?

Figures 2a and 3a: could the figure be modified to include better annotations for each cluster in the panel? It is difficult to look at the panel and have to refer back to the legend constantly.

Figure 2B: better choice of where to put the arrow heads could be made.

Figure 2C: I don't think some of those arrow heads are correct.... They don't seem to be pointing at the co-expressing cells?

Figure 2F,G: Can the authors clarify why Acta2 and Tagln antibodies are both used to label smooth muscle cells?

Reviewer #3 (Remarks to the Author):

De Bono and collaborators employ single-cell RNA sequencing (scRNA-seq) to (i) investigate developmental transitions that take place during crest cell (CNCC) development and (ii) examine the function of TBX transcription factors in this cell population.

The authors begin by characterizing the transcriptional profiles of Wnt1-Cre;EGFP positive cells from the pharyngeal region of E8.5, E9.5 and E10.5 embryos. They examine the transcriptional signature of the CNCC population at E9.5 and E10.5, which allows them to construct a map of CNCC fate that they validate through cell fate probability analysis using the CellRank pipeline. They subsequently perform scRNA-seq in wild-type and Tbx1 mutant embryos to identify differentially expressed genes. The results show that mutant embryos display upregulation of BMP genes and downregulation of FGF genes in the pharyngeal region. By analyzing data from Mesp1Cre control and Tbx1 conditional null embryos (at E9.5) with the CellChat software, the authors identify an increase of BMP signaling from the mesoderm to NCC in Tbx1 mutant embryos.

The datasets generated by the authors allow for the identification of a set of genes expressed during CNCC development. scRNA-seq of Tbx1 mutant embryos provides compelling evidence that loss of this TF results in the misregulation of the FGF and BMP signaling pathways. However, the study is limited by the strategy utilized to analyze the single cell data, which lacks precision. Instead of combining scRNA-seq datasets to delineate developmental trajectories (from 8dpc to 10.5dpc), data from each embryonic stage is analyzed separately. This limits insights into the developmental transitions that characterize CNCC development. In addition, the model proposed by the authors on the interplay between TBX transcription factors and FGF/BMP signaling is enticing but lacks further functional and biochemical validation. These shortcomings hinder the potential impact of the study.

Main concerns:

- The embryo dissections are arbitrary, which skews the proportions of cells that are being analyzed. The vast majority of the cells from 9.5dpc are cranial; PA1 gets dissected out in 10.5dpc, etc. While this does not compromise the analysis (as the authors can identify cNCCs), it does affect the quality of the datasets since the proportion of the populations of interest varies widely from stage to stage.

- Re: Figure 1. - "...and data from 36,721 NCCs were obtained" – were these GFP+ cells or NCCs? Wnt1-Cre labels a lot of cells that are not NCCs. Also, where are the CNS cells in the UMAP? Were these filtered out? Are the cells labeled as "neuronal/neural" supposed to be crest-derived neural cells or CNS cells?

- Re: Figure 1. - "Cluster C3 at E9.5 is similar to C2 at E10.5" –statement lacks explanation/quantification /statistics; it is not clear what the authors mean by this. Furthermore, the clustering assignment is based on a handful of genes and is not entirely convincing. Moreover, the authors have not included supplementary data on the control metrics that they used to determine the quality of their scRNA-seq dataset.

- The strategy to define cNCC gene signature in Fig 2 is confusing, as Acta2 is only expressed in a very small number of cells. What about markers that have been shown to be crucial for cNCC identity (e.g. TGIF, MAFB, etc)? Additionally, the genes of the "CNCC genetic signature" (Isl1, Gata, Hand) have all been reported before, so I am unsure about much novelty was uncovered by the scRNAseq.

- Figures 2 and 3 should be rearranged, as the text goes back and forth between them. Perhaps the results identifying and validating the CNCC gene signature for stages E9.5 and E10.5 (i.e., Figure 2A-G and Figure 3A-H) should be consolidated. Even better – the authors could combine the cNCCs of both stages (and perhaps include 8.5 dpc as well) to perform RNA velocity and inference of developmental trajectories. As it is, the datasets from different stages are underutilized.

- The rationale for selecting the genes in Figure 3L as lineage driver genes whose expression correlates with SM-CNCC fate acquisition is unclear from the main text/methods. These genes represent a relatively small subset of the complete list of genes that show similar expression patterns when cells are ordered by smooth muscle fate probabilities (Supplementary data 5).

- In the text (lines 219 – 221; re: Fig 5), the authors state that loss of Tbx1 results in the increased expression of genes downstream of BMP signaling in clusters 4 and 8. However, they only show graphs summarizing differential gene expression and GO term analysis for cluster 8. What happens in cluster 4? PA3 (represented at E9.5 in cluster 4) is of particular importance since CNCCs will begin entering the OFT from this PA later during E9.5 (as the authors have previously shown).

- The rationale for focusing on BMP and MAPK pathway-related genes for the differential gene expression analysis centers around the importance of these signaling pathways for NCC development and migration during embryogenesis (as stated by the authors in lines 24-226). Yet, the gene ontology analysis of upregulated genes in cluster 8 also reveals the enrichment of genes involved in the regulation of other signaling pathways (e.g. Wnt), which are also consequential for NCCs development. This should be clarified.

- The authors propose a model where mesodermal TBX1 promotes FGF signaling, which in turn inhibits BMP activity in CNCCs. The model is consistent with the results from the scRNA-seq analysis in the TBX1 mutants, but much of it seems speculative. For instance, I could not find data supporting that increase in BMP signaling in the mutants is due to the loss of FGF signaling. Epistatic experiments would be necessary to show that this is indeed the case (since other signaling pathways are also affected in the mutant). The study also lacks experiments that would show a direct interaction between the components of the model.

Response to the Reviewers. We thank the three reviewers for their insightful and thoughtful comments. We hope we have significantly improved the manuscript. We include a point by point response to each comment to each of the Reviewers, in **blue font**. We also provide a table here, that summarizes our changes.

Figures and Data	
New Figures	Original Figures
Figure 1: -added more details on scRNA-seq cluster annotations -added Twist1 RNAscope at E8.5, E9.5 and E10.5	Figure 1
Figure 2: -added Wnt1-Cre;EGFP and TAGLN immunostaining at E9.5 -indicated newly identified genes connected to CNCCs by asterisk (*)	Figure 2
Figure 3: -added Cnn1 and Rgs5 RNAscope -indicated newly identified genes connected to CNCCs by asterisk (*) -updated GO analysis in P	Figure 3
Figure 4: -added additional H & E staining showing aortic arch hypoplasia in Wnt1-Cre;Tbx2ff;Tbx3ff embryos (F,I,L) -updated the table in M -added IF image to show the range of expression of ACTA2 in PAA of Wnt1-Cre;Tbx2ff;Tbx3ff embryos at E11.5 (from almost no expression to low level expression)	Figure 4
Figure 5:-removed GO analysis of DEGs in C8 of E9.5 Control and Tbx1 null integrated scRNAseq data -added scatter plot of DEGs in cluster C4 of E9.5 integrated scRNAseq data -added violin plots for Msx2 , Bambi , Spry1 and Myc expression in Control and Tbx1 null embryo scRNA-seq data -added Phospho-ERK immunofluorescence at E9.5 in Control and Tbx1 null embryos.	Figure 5
Figure 6:-added RNAscope for Bmp4 in Control and Tbx1 null embryos at E9.5	Figure 6
Figure 7:-added violin plots for Msx2 , Bambi , Gata3 and Tbx2 in Control and Tbx1 null scRNAseq data at E10.5 -added GO analysis of upregulated genes in cluster C4 of integrated scRNA-seq data at E10.5	Figure 7
Figure 8:-updated models in A and B	Figure 8
Supplementary Figures	
New Supplementary Figures	Original Supplementary Figures
Supplementary Fig. 1 (Cluster identification of single cell RNA-seq data of NCCs at E8.5.)	
Supplementary Fig. 2 (Cluster identification of single cell RNA-seq data of NCCs at E9.5.)	
Supplementary Fig. 3 (Cluster identification of single cell RNA-seq data of NCCs at E10.5.)	
Supplementary Fig.4	Supplementary Fig.1
Supplementary Fig.5 (Distinction between cardiac skeletal muscle progenitor cells and CNCCs in the OFT expressing ACTA2.)	

Supplementary Fig.6 (Additional feature plots for genes identified in analysis of transcriptional dynamics of cardiac NCCs at E9.5 and E10.5.)	
Supplementary Fig.7 (Cardiac smooth muscle cells fate trajectory of integrated scRNA-seq datasets of NCCs from control embryos at E8.5, E9.5 and E10.5.)	
Supplementary Fig.8 (Reduction of the thickness and number of smooth muscle layers in the aberrant right subclavian artery of Wnt1-Cre;Tbx2ff;Tbx3ff embryos at E15.5.)	
Supplementary Fig.9	Supplementary Fig.2
Supplementary Fig.10	Supplementary Fig.3
Supplementary Fig.11 (Clustered cells in scRNA-seq data from control embryos are also clustered together in integrated control and Tbx1 null scRNA-seq data analysis at E9.5.)	
Supplementary Fig.12	Supplementary Fig.4
Supplementary Fig.13 (Bmp7 expression is strongly maintained in the pharyngeal apparatus of Tbx1 null embryos at E9.5.)	
Supplementary Fig.14 (Clustered cells in scRNA-seq data from control embryos are also clustered together in integrated scRNA-seq data from control and Tbx1 null embryos at E10.5.)	
Supplementary data	
New Supplementary data	Original Supplementary data
Supplementary data 1	Supplementary data 1
Supplementary data 2	Supplementary data 2
Supplementary data 3	Supplementary data 3
Supplementary data 4	Supplementary data 4
Supplementary data 5	Supplementary data 5
Supplementary data 6	Supplementary data 6
Supplementary data 7	Supplementary data 7
Supplementary data 8	Supplementary data 10
Supplementary data 9: Gene ontology biological processes of upregulated genes in pharyngeal NCCs (C4 in Figure 7) from control and Tbx1 null embryos at E10.5.	
Supplementary data 10	Supplementary data 11
Supplementary data 11	Supplementary data 12

REVIEWER COMMENTS

Reviewer #1 (Remarks to the Author):

In this manuscript, De Bono and colleagues investigate single-cell transcriptomic profiles of neural crest cells from the pharyngeal apparatus to the heart in normal and *Tbx1*-null embryos. They demonstrate transitions of NCC differentiation stages characterized by different transcription factors, which are

affected non-autonomously by loss of Tbx1. Among affected transcription factors, co-inactivation of Tbx2 and Tbx3 caused aberrant right subclavian artery with incomplete penetrance. Transcriptomic profiles further showed upregulation of BMP signaling and reduced MAPK signaling in CNCCs of Tbx1-null embryos, which lead them to argue that alteration of mesoderm-derived FGF signaling may result in excessive BMP activation responsible for cardiac malformations in 22q11.2 deletion syndrome. Overall, data are generally of high quality and are well presented. However, the data are largely descriptive and correlative in nature and the mechanistic novelty is limited and lacking direct validation especially for the source of responsible BMP ligands and the involvement of altered BMP and MAPK signaling pathways in the Tbx1-null phenotype.

Major concern: The data are largely descriptive and correlative in nature and the mechanistic novelty is limited and lacking direct validation especially for the source of responsible BMP ligands and the involvement of altered BMP and MAPK signaling pathways in the Tbx1-null phenotype.

We used the single cell RNA-sequencing technology with RNAscope and immunofluorescence validation to understand how NCCs transition to early smooth muscle cells in the normal state. Our mechanistic novelties are 1) we identified the gene signature of CNCCs in mammals; 2) showed that *Tbx2/3* function in CNCC fate progression; 3) how cell fate progression and cell types are altered when we inactivated *Tbx1*; 4) we uncovered new genes, not known for CNCCs and their transitions; 5) further validation with molecular markers (*Bmp4*, *Bmp7*, *Twist1*); 6) immunofluorescence of pSMAD and pERK to validate alteration of BMP and MAPK signaling. We took a genetic approach and identified genes that are novel and known for transition to different states.

Specific comments:

1. Regarding the single-cell transcriptomic analysis, Chen et al. have published a paper showing partially overlapping data of CNCCs with diverse lineages (EMBO rep. 2021), which should be referred to in the manuscript and discussed together.

In Chen et al., EMBO, 2021, the authors focused on CNCC differentiation from E10.5-P7, once *Wnt1-Cre* derived cells have migrated to their final destinies in the heart. This study did not include analysis of the progenitor cells in the pharyngeal arches. In the Chen paper, the transitions of CNCCs were at more mature differentiation states than in this study.

We now state on lines, 79-82, "In another recent study using mouse scRNA-seq data, Chen and colleagues focus on CNCC differentiation from E10.5 to P (postnatal day) 7, once *Wnt1-Cre* derived cells have migrated to their final destinies in the heart¹⁶. However, this study did not include analysis of the cardiac progenitor cells in the pharyngeal arches."

2. The authors characterize *Acta2*-expressing cells as smooth muscle cells. However, *Acta2* is also expressed in different cell types such as myofibroblasts, pericytes and muscle progenitors during embryonic development. Indeed, *Acta2* expression is found in NCCs migrating into the endocardial cushion at E10.5 (Fig. 3F), whose phenotype is likely immature mesenchyme. The authors should characterize *Acta2*-expressing cells more definitely with several cell type markers.

Thank you for your comment. We realize that there are few specific smooth muscle gene markers expressed at E9.5 and E10.5. We used *Acta2* as a marker, because it is one of the earliest markers of smooth muscle cells, but we are aware that *Acta2* is not specific for smooth muscle cells only, as indicated by the Reviewer. Accordingly, we have clarified that the *Acta2*⁺ cluster more likely represents progenitors

of smooth muscle cells and perhaps other mural cells like pericytes. Unfortunately, progenitors of smooth muscle cells do not express characteristic mature smooth muscle markers such as *Myh11* (*Myh11* is a marker used in Chen et al, 2021), however there were very few cells expressing it in our hands at E10.5; see feature plot below. We now include new immunostaining and RNAscope *in situ* hybridization (Figure 2 and 3) data to better characterize the smooth muscle progenitor cells in the outflow tract. We used *Tagln* (Transgelin), *Cnn1* (Calponin) and *Rgs5* (Regulator of G Protein Signaling 5). CNN1 is a basic smooth muscle protein, RGS5 is expressed in vascular smooth muscle, TAGLN is ubiquitously expressed in vascular smooth muscle cells and is an early marker of smooth muscle differentiation. RGS5 is a marker for both smooth muscle and pericytes, so we used all these markers to help identify smooth muscle progenitor cells. We also include a sentence that lists additional markers of smooth muscle progenitors in the *Acta2* expressing cell cluster, from Supplementary data 3. This is spelled out in lines, 132-135, it now reads, “*Acta2* is a representative marker gene of progenitor smooth muscle cells, but it is not exclusive of smooth muscle cell types. Additional early markers of smooth muscle cells in addition to *Acta2* are *Tagln*, *MyI9*, *Myh9*, *Rgs5* and *Cnn1*²⁵⁻²⁷.”.

By combining *Wnt1-Cre;GFP* lineage tracing experiments and IF/RNAscope of the aforementioned genes, we refer these cells, including *Acta2* expressing cells as SM-CNCCs (*Acta2*+ NCCs). This is now indicated in the text on lines, 161-163, “CNCCs of the OFT derived from the *Wnt1-Cre;GFP* genetic lineage and expressing *Acta2*, *Tagln*, *Cnn1*, *Rgs5*, *Isl1* and *Gata3* are smooth muscle progenitors.” and Figure 2 and 3.

Furthermore, in order to discriminate cardiac skeletal muscle progenitor cells expressing *Acta2* from NCCs expressing *Acta2*, we performed co-immunostaining at E10.5 for *Wnt1-Cre;GFP*; ACTA2 and MF20 (Myosin Heavy Chain Antibody) (Supplementary Figure 5). MF20 was detected in cardiac skeletal muscle but not smooth muscle cells. Data show the absence of MF20 detection in NCCs contributing to the cardiac outflow tract. This is now indicated in the text lines, 144-147, “Furthermore, we performed co-immunostaining at E10.5 for *Wnt1-Cre;GFP*, ACTA2 and MF20 (Myosin Heavy Chain Antibody) to discriminate cardiac skeletal muscle cells, that also express ACTA2, from adjacent SM-CNCCs in the OFT.

Supplementary Fig. 5 shows that MF20 is detected, as expected, in cardiac skeletal muscle cells but not in SM-CNCCs of the OFT.”.

3. In the PHATE map of E9.5 CNCCs (Fig. 2H), clusters are allocated to different regions. For example, C5 and C3 likely correspond to PA2 and PA3, respectively. Then, how did the authors construct the heatmap in Fig. 2K along the latent time? Is it derived from a mixture of data from spatially different clusters? If so, how is it validated? There may be a similar concern for the analysis shown in Fig. 3, where the clusters are not spatially allocated but different Hox gene expression patterns indicate the clusters might be regionally distinct cell populations.

The PHATE map can capture the gene expression relationship between cells, better than UMAP or tSNE maps, both within and between clusters. In particular, the cluster position in a UMAP or tSNE plot does not always represent the true differentiation relationship between clusters. Thus, we use PHATE mapping in Figure 2H. Nevertheless, it’s computed from gene expression similarity. To infer cell differentiation relationships like trajectory, CellRank is a better software because it combines gene expression similarity and RNA velocity, which can predict the differentiation trend of a cell. The latent time (pseudotime) in the heatmap of Fig. 2K is from CellRank software, independently from the PHATE map method, although the same colors were used. The PHATE map of latent time shows the cell fate probability to become SM-CNCCs, mostly to help see the relationship between clusters. Latent time is a continuous variable, and the Reviewer is correct that cells from different clusters were analyzed together and some cells in different clusters may have the same latent time, especially for cells in early differentiation states. It is important to note that cells with small (i.e., early) latent time values simply suggest they are not committed to the smooth muscle fate and can become other cell types later, for examples, for the cells expressing *Sox10*. From that perspective, more attention should go to cells with high latent pseudotime (right side of the heatmap). Additionally, there are other terminal states in our CellRank analysis, but our focus here is the cell probability leading to the SM-CNCC *Acta2+* fate.

Several of the genes in the heatmap have been validated by us using RNAscope or IF (*Isl1*, *Gata3*, *Msx2*, *Bambi*, *Tbx2* and *Tbx3* in Fig. 2, 3 and 5), and many have been validated in the literature (PMID: 19008477; PMID:25363691; PMID:23011393). Their expression patterns are consistent with our cell fate inference.

4. About 40% of *Tbx2/3* cKO embryos have an aberrant right subclavian artery but no other defects. Based on immunostaining shown in Fig. 4J-L, the authors discussed the role of *Tbx2/3* in smooth muscle differentiation. Is the failed smooth muscle differentiation of the right fourth PAAs due to defects in differentiation capacity or delayed NCC deployment and/or differentiation? Are the aberrant right subclavian arteries defective for the smooth muscle layer?

Based on our immunostaining at E11.5, it appears that CNCCs contribute normally to the PAAs of *Wnt1-Cre;Tbx2^{f/f} Tbx3^{f/f}* embryos (figure 4 N,O,P), ruling out a possible defect of deployment of NCCs to the PAAs. This is stated on lines 253-256, and it reads, “By immunostaining on coronal sections of *Wnt1-Cre/+;Tbx2^{f/f};Tbx3^{f/f};ROSA-EGFP^{f/+}* embryos at E11.5 using GFP and ACTA2 antibodies, we found that NCCs contributed normally to the right and left 4th PAAs but partially or completely failed to differentiate into smooth muscle cells (Fig. 4N-Q).”.

In addition, we noticed variability in expression of *Acta2* in PAAs of *Wnt1-Cre;Tbx2^{f/f} Tbx3^{f/f}* embryos at E11.5, ranging from a total absence to reduced expression. Figure 4 has been updated accordingly by adding a panel that expression is variably reduced. This could explain the basis for the partial penetrance of the aberrant right subclavian artery (ARSA) in mutant embryos at E15.5.

We qualitatively analyzed control versus *Wnt1-Cre;Tbx2f/f;Tbx3f/f* embryos at E15.5 to compare to the thickness of the muscular wall with/without aberrant right subclavian artery (Supplemental Figure 8). We found reduced thickness and reduced number of muscle layers in the aberrant right subclavian arteries of *Wnt1-Cre;Tbx2f/f;Tbx3f/f* embryos (n=3) suggesting an absence of the smooth muscle layer, consequently to a possible failure of smooth muscle differentiation.

The pictures also show low *Acta2* signals in the left PAAs as compared with the right PAAs. Were no defects observed in the left fourth PAA-derived aortic arch?

We reanalyzed the histological sections and observed hypoplasia of the aortic arch (derived from the left PAAs) in 25% (2/8) of the *Wnt1-Cre;Tbx2f/f;Tbx3f/f* mutant embryos at E15.5. Figure 4 has been updated accordingly.

5. How about the ductus arteriosus derived from the left sixth PAA?

No ductus arteriosus defects were observed in *Wnt1-Cre;Tbx2f/f;Tbx3f/f* embryos.

6. Regarding the analysis on *Tbx1* mutants, presented data clearly demonstrate differentially expressed genes between wild-type and *Tbx1*-null mutant embryos, but mechanistic relationships are speculative. Although several ligand-receptor pairs were extracted as altered signals in *Tbx1* mutants, direct evidence is lacking as for their contribution to the *Tbx1*-null phenotype. Rescues of the *Tbx1*-null phenotype by conditional genetic inactivation or activation of BMP or MAPK signaling pathways are expected to give a direct support for their contribution. It is also important to identify the source of BMP ligands responsible for the *Tbx1*-null phenotype.

We agree with the comments of the Reviewer. There are 2 main signaling pathways that we focused on in this manuscript, because they were the most changed from our single cell RNA-sequencing data, and these are the FGF and BMP pathways. It is well known in the literature that loss of *Tbx1* affects expression of FGF ligands, including FGF8 and FGF10 (PMID: 17238155; PMID: 12223416; PMID: 17000704). FGFs act via the MAPK signaling pathway. We found, and now validated that there is reduced MAPK kinase signaling in NCCs (clusters C4 and C8). We added immunostaining for p-ERK (readout of MAPK signaling pathway) in Figure 5. This confirms that there is reduction of MAPK signaling in NCCs in the absence of *Tbx1* at E9.5. This provides mechanistic insights that altered FGFs act via MAPK signaling in NCCs.

In the original text, we found an increase in BMP downstream genes (e.g. *Msx1*, *Msx2*, *Id1*, *Bambi*, etc) with wider expression of p-SMAD signals in *Tbx1* null mutant embryos. But the source of the BMP ligands that could mediate this was not described in the text. We now performed RNAscope *in situ* hybridization experiments to detect expression of *Bmp4* and *Bmp7* in control and *Wnt1-Cre; Tbx1*^{-/-} embryos at E9.5. We found that both genes continue to be robustly expressed in the pharyngeal mesoderm and endoderm, respectively of control and mutant embryos. These tissues directly interact with adjacent NCCs in the pharyngeal region. This is now shown in an update of Figure 6 and in Supplementary Figure 13. Based upon this, it is possible that these two ligands, along with changes in expression of other genes in the pathway, could be involved in premature activation and increased expression of BMP downstream genes observed.

Both increase in BMP pathway (PMID: 32105214) and decrease in the MAPK pathway (PMID: 18952847) in NCCs are separately, associated with developmental defects of NCCs leading to cardiac outflow tract malformations. This is now added on line 300-302 “Signaling by BMP⁴¹ and growth factors activating the

MAPK pathway^{42,43} are two signaling pathways known to be critical in NCCs for cardiac development.” and lines 455-456, “It is known that loss of FGF ligands indirectly results in NCC mediated cardiovascular anomalies^{42,59,60}.”

We tested for genetic rescue of cardiac outflow tract defect by reducing BMP signaling in NCCs in *Tbx1* null embryos. We performed the following genetic cross:

Males that are *Wnt1Cre;Tbx1+/-;Smad1f/+* were crossed with females that are *Tbx1+/-; Smad1f/f*. The goal was to inactivate one or both alleles of *Smad1* in NCCs in *Tbx1*^{-/-} null mutant embryos. Embryos were collected at E15.5 to look for phenotypic evidence of genetic rescue.

Results:

Genotype	Number of analyzed embryos	Persistent truncus arteriosus	Ventricular septal defect	Double outlet right ventricle
Wnt1-Cre; Smad1f/f	2	0%	0%	0%
Wnt1cre; Tbx1+/-; Smad1f/f	9	0%	22% (2)	11% (1)
Wnt1-Cre;Tbx1+/-;Smad1f/+	2	0%	0%	0%
Wnt1-Cre;Tbx1-/-;Smad1f/f	4	100% (4)	100% (4)	0%
Wnt1-Cre;Tbx1-/-;Smad1f/+	2	100% (2)	100% (2)	0%

Tbx1^{+/-} embryos in the mixed genetic background that was used do not have aortic arch nor heart defects. Our results indicate that there was no genetic rescue in *Wnt1-Cre;Tbx1-/-;Smad1f/f* nor *Wnt1-Cre;Smad1f/+* embryos. Moreover, some embryos with these genotypes had a more severe phenotype (we didn't include in the table the embryos that were already dying or were dead and that appeared smaller than controls) than that observed in *Tbx1*^{-/-} embryos. *Tbx1*^{-/-} embryos die at birth. Some of the mutants with inactivation of *Smad1* showed lethality at E12.5-E13.5. The phenotypes were more severe rather than being less severe.

Possible explanations for our observed results are that:

- Rescue of NCCs may not be sufficient to rescue the *Tbx1* null phenotype because *Tbx1* also plays a critical role in the development of the pharyngeal mesoderm, including the second heart field, and pharyngeal endoderm, which are both essential for correct cardiac outflow tract development.

- BMP is essential for generation of NCCs in the dorsal neural tube. By altering the BMP signaling pathway in the *Wnt1-Cre* lineage, production of NCCs in the neural tube may be affected.
- Likely many signaling pathways in addition to BMP signaling are important downstream of *Tbx1* for formation of the aortic arch and cardiac outflow tract.

We did not include this experiment in the manuscript because it does not provide any new insights.

7. What is the relationship between clusters shown in Fig 1E and Fig 5B? Then, is it possible that altered BMP and MAPK signaling pathways affects transcriptomic profiles of each CNCC cluster?

To address this concern, we created a Supplementary Figure 11. This figure includes a table that shows the absolute number of cells from control embryos (based on actual barcodes) in each cluster in Figure 1E and Figure 5B, thus directly relating the cluster assignment in the two figures. We found that most of the cells in a cluster identified in our analysis of E9.5 control sample stayed together as a cluster in our integrated analysis of data from control and *Tbx1* null samples. This indicates that the overall transcriptional profiles of the CNCCs are similar between control and *Tbx1* null embryos. This is now described on lines 280-284 and it now reads, "We confirmed that most of the cells in each cluster of the integrated control and *Tbx1* null scRNA-seq dataset (shown in Fig 5B) were also clustered together in the individual datasets (Fig. 1E; Supplementary Fig. 11). This indicates that although affected, the general transcriptional profiles of the CNCCs are still conserved between control and *Tbx1* null embryos."

We analyzed differentially expressed genes between control and *Tbx1* null embryos in each cluster (C1, C3, C4, C5 and C9 in Fig. 5B and Supplementary data 7). We mainly observed altered BMP and MAPK signaling in clusters C4 and C8 as shown in Fig. 5D-H.

We additionally created a Supplementary Figure 14 including a table showing that most of the clustered cells in control scRNA-seq data analysis at E10.5 (Figure 1) are also clustered together in integrated control and *Tbx1* null scRNA-seq data analysis at E10.5 (Figure 7).

Reviewer #2 (Remarks to the Author):

This is a detailed and comprehensive paper describing the transcriptome of cardiac neural crest cells that migrate through the pharyngeal region and to the outflow tract during a critical time of cardiovascular development in the mouse. The transcriptome of neural crest cells from *Tbx1*-null embryos has also been analysed and key gene expression differences identified.

There are, however, some issues that need clarification.

The introduction should briefly describe the formation and remodelling of the PAA, particularly as this information is required to understand the ARSA phenotype described later. Also, describe the role of NCC in outflow tract septation as this is not restricted to smooth muscle formation.

More information has been provided in the Introduction. It now reads: "The pharyngeal apparatus is a dynamic embryonic structure consisting of individual pharyngeal arches (PA), forming in a rostral to caudal manner from mouse embryonic day (E) 8 to E10.5. A subset of pharyngeal NCCs migrate through the caudal pharyngeal arches, PA3-6, and surround the pharyngeal arch arteries (PAAs), while others continue to migrate to the cardiac outflow tract (OFT), both differentiating to vascular smooth muscle cells². The PAAs form in a similar, rostral to caudal manner and are linked continuously with the cardiac OFT. The PAAs become remodeled to form the aortic arch and vascular branches that include the subclavian

arteries. The 4th PAAs are particularly important in that the left 4th PAA is critical to form the aortic arch with the left subclavian artery and the right 4th PAA forms the branch to the right subclavian artery. Differentiation of CNCCs into smooth muscle cells of the PAAs is important for remodeling of the PAA². The OFT elongates by addition of second heart field mesoderm cells and this is required for looping and septation of the OFT. It has been shown that CNCCs are required for addition of second heart field mesoderm cells to expand the length of the OFT³. In addition, CNCCs are required for OFT septation by invading and allowing the endocardial cushions of the OFT to come into direct cell-cell contact, fuse and form the aorticopulmonary septum^{2,4,5}. Complete ablation of NCCs from PA3-6 results in interruption of the aortic arch and arterial branching defects as well as persistent truncus arteriosus of the OFT⁶.”

Based on the data acquired from control embryos, the authors tested the role of Tbx2 and Tbx3 in cardiovascular development by conditionally deleting both genes simultaneously from NCC. A proportion of mutant fetuses have ARSA.

Line 331: mentions ‘reduced penetrance’. What is this compared to? Or should this be referring to a low penetrance?

The wording has been changed to “partially penetrant”.

The legend for Figure 4 states that there is a “...strong reduction of Acta2 expression in the right 4th PAA...in mutant embryos...”. The implication is that it is the right PAA only that is lacking smooth muscle cell investment which causes the ARSA phenotype. Panel K and L focus on the right 4th PAA in mutants, but the left PAA (which is phenotypically unaffected at E15.5) looks to have a similar level of smooth muscle staining which would suggest that the effect (if any) is symmetrical and equal for each 4th PAA.

We reanalyzed the histological sections and observed hypoplasia of the aortic arch (derived from the left PAAs) in 25% (2/8) of the *Wnt1-Cre;Tbx2f/f;Tbx3f/f* mutant embryos at E15.5. Figure 4 has been updated accordingly.

How many mutant embryos were analysed? Were they stage-matched with the controls? Methods suggest at least 3 were analysed but this needs to be clarified, particularly as only 38.5% of fetuses develop the ARSA phenotype.

We analyzed 8 controls and 8 *Wnt1-Cre;Tbx2f/f;Tbx3f/f* embryos of the same developmental stage. Control and *Wnt1-Cre;Tbx2f/f;Tbx3f/f* mutant embryos from the same litter were compared. We added the numbers to the text.

If all embryos have reduced 4th PAA smooth muscle staining, why do the majority of fetuses have a normal right subclavian artery? The coronal sections suggest that 4th PAA formation occurs – the vessel is well formed - so can the authors please discuss what they think happens to this vessel? For example, do they suspect that a lack of smooth muscle cell investment causes the remodelling vessel to collapse, resulting in ARSA?

Overall, this part of the study needs more discussion. E.g. why is there a low penetrance of 4th PAA defects, why is the right 4th PAA affected and not the left, what happens to the vessel if there are fewer smooth muscle cells, etc.

All analyzed *Wnt1-Cre;Tbx2f/f;Tbx3f/f* mutant embryos have reduced 4th PAA smooth muscle staining but we noticed variability in expression of *Acta2* in PAAs of *Wnt1-Cre;Tbx2f/f;Tbx3f/f* embryos at E11.5,

ranging from a total absence to reduced expression. Figure 4 has been updated accordingly to show this variability. This could explain the basis for the partial penetrance of the aberrant right subclavian artery (ARSA) in mutant embryos at E15.5.

In addition, several studies show that abnormal 4th PAAs at early/mid-embryogenesis have the ability to recover over time in embryogenesis and contribute normally to the formation of the aortic arch and aortic arch branching (PMID: 11242049; PMID: 11971873; PMID: 19700621; PMID: 20110535; PMID: 31444215; PMID: 33249995).

We also qualitatively analyzed the thickness of the muscular wall of the normal and aberrant right subclavian arteries in E15.5 *Wnt1-Cre;Tbx2ff;Tbx3ff* embryos compared to controls (Supplementary Figure 8). We found reduced thickness and reduced number of muscle layers in the aberrant right subclavian arteries of *Wnt1-Cre;Tbx2ff;Tbx3ff* embryos suggesting an absence of the smooth muscle layer, consequently to a possible failure of smooth muscle differentiation.

Although this part of the study is independent of the *Tbx1* analysis, this group has previously published that *Tbx1*-hets on a Swiss-Webster background do not have 4th PAA defects, unlike on other backgrounds such as C57Bl/6. Could the low penetrance of 4th PAA defects in the *Tbx2/3* mutants be related to genetic background?

A mixed genetic background was used for these experiments. It is possible that the penetrance of the defects observed is related to differences in genetic background. We addressed this point, by adding the following sentence on lines 422-424, "Partially penetrant hypoplasia of the aortic arch and arterial branching defects of *Wnt1-Cre;Tbx2^{ff};Tbx3^{ff}* embryos can result from the ability of the PAAs to recover during development^{13,61-65}". Another reason for partial penetrance in mutant embryos could be because mice with a mixed genetic background were used in this study."

Line 513: Suppl Table 1: why are the E10.5 stages split into two groups (there are two sets of control and two sets of mutant)?

We performed biological replicates (#1 and #2) for E10.5 of control (CTRL) and *Tbx1* null mutant embryos. This is now clarified in the text below the table.

Were these samples pooled for analysis?

We only pooled data from different samples for integrated analysis of CTRL vs *Tbx1* null mutant embryos in Figure 6.

Suppl Table 1 suggests 20 embryos were used altogether. The manuscript says that 36,721 NCCs were sequenced (line 85). This doesn't match up with the data from the 20 embryos in Suppl Table 1 which says a total of 65,852 NCCs were sequenced.

The number, 36,721 is the number of sequenced NCCs only for control embryos at E8.5, E9.5 and the two replicates at E10.5 for Figure 1.

The number of 65,852 NCCs correspond to the total number of embryos (controls + *Tbx1* null mutant embryos at all stages examined)

This is now clarified in the text and lines 104-106 reads, “EGFP positive NCCs were purified by FACS and the Chromium 10X platform was used to perform scRNA-seq and data from 36,721 NCCs of control embryos were obtained (Supplementary Table 1).”.

Line 206-209: Why are there fewer NCC in *Tbx1*-nulls? Conversely, Figure 5D shows that there more NCC in PA1-2 in *Tbx1*-nulls compared to controls. Is this related to the failure in pharyngeal segmentation preventing NCC migration, or do fewer NCC cause the segmentation defect? Can the authors speculate why this might be?

The first question is why are there fewer NCCs in *Tbx1* nulls. Our data shows that the relative proportion of NCCs is normal in the neural tube and in the *Sox10* expressing population. Thus, NCCs are produced normally in the neural tube and migrate normally to some extent. But, within the pharyngeal region, there is misrouting of NCCs to PA1 (and part of PA2; PMID: 11971873), and fewer NCCs in the caudal pharyngeal region. We suggest that this is due to both failed signals from *Tbx1* expressing cells to NCCs and failure of segmentation of the caudal pharyngeal arches.

A previous study by Veitch et al., 1999, showed that after ablation of NCCs in the chick, pharyngeal segmentation can take place (PMID: 10607595). Based upon the Veitch study we don't think that fewer NCCs cause the segmentation defect in *Tbx1* null embryos. *Tbx1* is expressed in the pharyngeal mesoderm and endoderm; inactivation in either tissue disrupts both signaling and segmentation. For example, Dr. Basson and colleagues found that loss of *Tbx1* in the endoderm results in failed segmentation of the caudal arches with fewer NCCs (PMID:24812002). In addition, *Tbx1* expression in the mesoderm is necessary for pharyngeal arch formation and segmentation of the pharyngeal apparatus, with fewer NCCs (PMID:16914493). Thus, it is possible that disruption of non-autonomous signaling from *Tbx1* expressing cells cause failed NCC fate progression and, separately, failed segmentation of the pharyngeal arches. It is possible that these are different consequences of loss of *Tbx1*.

In the Discussion we now added this in the text on lines 493-507, “One of the reasons for disruption of NCCs in *Tbx1* mutant embryos is failed segmentation of the caudal pharyngeal arches by the endoderm. Alternatively, disruption of NCCs could affect the pharyngeal endoderm resulting in failed segmentation. A previous study by Veitch and colleagues showed that after ablation of NCCs in the chick embryo, pharyngeal segmentation still took place⁷⁸. This suggests that pharyngeal segmentation defects in *Tbx1* null embryos is not due to reduced number or defective NCCs. There are many reasons for failed segmentation of the caudal pharyngeal apparatus due to *Tbx1* inactivation. The tissue specific functions of *Tbx1* in this process has been evaluated. Loss of *Tbx1* in the pharyngeal endoderm autonomously affects segmentation of the caudal pharyngeal apparatus leading to reduced contribution of CNCCs⁷⁹. The pharyngeal endoderm is also an important source of signaling during development, including FGF ligands⁵⁹, that could potentially affect NCCs development. It was found that *Tbx1* expression in the mesoderm is necessary, non-autonomously for pharyngeal segmentation⁴⁵, suggesting that defects in signaling from the adjacent mesoderm that expresses *Tbx1*, causes failed segmentation. Loss of *Tbx1* in the mesoderm also leads to failed CNCC migration and fate progression.”.

Given that *Tbx1* is expressed in ectoderm, endoderm and the mesoderm of the pharyngeal arches, signalling from one or more of these tissues must be influencing NCC behaviour. The discussion only briefly mentions the possibility that the endoderm may also be involved in *Tbx1* signalling so this could be expanded on to include the ectoderm. Mutant mouse studies have been published to show that *Tbx1* genetically interacts with other genes in these tissues to regulate PAA morphogenesis.

We have now expanded the statement on line 502-509, that now reads, “The pharyngeal endoderm is also an important source of signaling during development, including FGF ligands⁵⁹, that could potentially affect NCCs development. It was found that *Tbx1* expression in the mesoderm is necessary, non-autonomously for pharyngeal segmentation⁴⁵, suggesting that defects in signaling from the adjacent mesoderm that expresses *Tbx1*, causes failed segmentation. Loss of *Tbx1* in the mesoderm also leads to failed CNCC migration and fate progression. The *Tbx1* expression domain in the pharyngeal ectoderm also serves as a signaling center for non-autonomous signaling^{64,80}. It will be interesting to evaluate how the exchange of signaling between the pharyngeal epithelia and NCCs are affected in the absence of *Tbx1*.”.

Line 199 Authors state ‘We found that *Tbx1* was not noticeably expressed in NCCs’. However, there are clearly some NCCs positive for *Tbx1* at E8.5 in Suppl. Fig 2 and some at E9.5. Although the authors present data that shows deleting *Tbx1* in NCC does not lead to cardiovascular defects, these few groups of double positive cells should be labelled, despite their being a small number.

After analyzing several sections, we confirm that *TBX1* is not noticeably expressed in NCCs. We apologize for the yellow staining in Supplementary Figure 9 (previously Supplementary Figure 2). This staining corresponds to auto fluorescent blood cells.

We did not detect *TBX1* expression in NCCs and we didn’t observe aortic arch or heart defects when we used *Wnt1-Cre* to inactivate *Tbx1* (Supplementary Figure 10). The same results were found in previous studies by other groups (PMID: 11971873). It is possible that there is some low level of *Tbx1* expression in NCCs, that wasn’t detected by RNAscope or immunofluorescence. It should be noted that inactivation of *Tbx1* in the *Wnt1-Cre* lineage leads to an absence of formation of the hyoid bone and postnatal lethality, suggesting the existence of some NCCs expressing *Tbx1* (PMID: 25209980). Thus, there must be specific expression in craniofacial skeletal progenitors in the third pharyngeal arch that we did not detect in our assays.

Why is there no supporting RNA scope for *Twist1* in Figure 1? Figure 1G-I focuses on the decrease in *Sox10* expression from E8.5-E10.5 and the simultaneous increase in *Twist1* expression, however, only *Sox10* RNA expression is validated with the RNA scope. As this is an important correlation *Twist1* RNA should also be validated/visualised in this way. Just as *Hox2a* and *3b* have also been validated in this Figure.

We now included *Twist1* RNAscope *in situ* hybridization at E8.5, E9.5 and E10.5 in Figure 1. This figure shows strong *Twist1* expression in the pharyngeal arches, but, as anticipated, little expression was observed in the neural tube or in early migrating cells to the pharyngeal arches.

Minor comments

Line 63: “*Est1*” should be “*Ets1*”

This is corrected.

Line 154: states ‘This data also indicates that a small fraction of P-CNCCs may directly differentiate to smooth muscle cells (blue fraction in the pie chart in C2)’. This reads as though the authors are suggesting some cells in cluster 2 differentiate directly to cells in cluster 10, an arrow depicting this should be included in Figure 3K. If the authors are referring to cells in C2 directly differentiating to cells in C4 then this isn’t clear and should be reworded in the manuscript.

We added additional information in the text and it reads on lines 203-206 “This data also indicates that a small fraction of CP-NCCs may directly differentiate to smooth muscle cells (blue fraction in the pie chart in C2 indicates high probability of a fraction of cells in C2 to transition directly to C10), in agreement with *Tbx2* and *Tbx3* expression in PAA-CNCCs at E10.5 (Fig. 3F, H, I)” We cannot add an arrow because the arrows in the figure were identified statistically.

Line 170: authors state ‘addition, our functional enrichment analysis identified genes associated with congenital heart disease such as tetralogy of Fallot, double outlet right ventricle and ventricular septal defects (Supplementary data 6)’. Looking at this data, it was in group 4 only of the segregated NCCs where this was identified, therefore this should be clarified in the manuscript.

We now specified in the text that this was identified only in group 4 (the most differentiated progenitor cells of smooth muscle). On lines 227-231, it reads “In addition, our functional enrichment analysis of the fourth, most differentiated set, identified genes associated with congenital heart disease (*Gata4/6*, *Hand1/2*, *Tbx20*, *Nrp1*, *Acta2*) such as tetralogy of Fallot, double outlet right ventricle and ventricular septal defects in the most differentiated cells (Supplementary data 6), which supports the importance of the genetic program of CNCCs in OFT formation and disease”.

Line 269: Authors state ‘Two additional ligand genes, *Edn3* and *Pdgfa* were not investigated regarding *Tbx1*’ which reads as if the authors chose not to investigate these and should be written as ‘To our knowledge there is no record of *Edn3* and *Pdgfa* having been investigated in relation to *Tbx1*’.

This has been fixed in the text.

Line 402: could be written more clearly, using the term recombined.

This line now reads, “After recombination of the genomic sequences between the LoxP sites, the reading frame and T-Box domain of *Tbx2* and *Tbx3* are both disrupted”.

Line 404: should be C57BL/6

This has been fixed.

Line 487: Refer to Suppl Table 1 as this contains the details of the number of embryos collected for analysis.

This has been corrected.

Line 498: More detail required for how the FACS was performed.

We now provide more detail. This line now reads, “Cells were FACS (Fluorescence Activated Cell Sorting) purified using a BD FACSAria II system in order to collect EGFP positive and DAPI negative living cells in 1x PBS containing 10% FBS”.

Line 499: ‘DAPI negative’ should be ‘DAPI positive’

It should be DAPI negative. DAPI negative cells correspond to living cells.

Line 533: Number of samples – implies that n=1 at E9.5 and n=2 at E10.5 were analysed. This doesn't fit with Suppl Table 1 details.

We performed 2 biological replicates at E10.5 and one at E9.5. Each sample that was submitted for scRNA-seq was a pool of 2 or more embryos of the same genotype at the same stage. This is now written in the text of the Methods, on lines 651-654 as “One scRNA-seq experiment was performed for control and *Tbx1* null embryos at E9.5 and two biological replicates were performed for control and *Tbx1* null embryos at E10.5. Each sample that was submitted for scRNA-seq was a pool of two or more embryos of the same genotype (Supplementary Table 1).”.

Line 823: Cluster C5 and C9 not labelled in Figure 5B, but these are shown in 5D

These labels have now been added.

Line 137 referring to Figure 2G: The authors state ‘the first CNCCs from PA3 were entering the OFT (Fig. 2G).’ However, Fig 2F does look to show some CNCCs in the OFT coming from PA3. The authors should clarify this statement to say there are a small number of early CNCC in the OFT from PA3, however, the stream of CNCCs from PA3 is fully established in 2G.

In Figure 2G, left, the cells entering the OFT are from PA2, which is highlighted with the arrow pointing to the region that PA2 connects to the OFT. Whereas in Figure 2G on the right, the OFT can clearly be seen connecting to PA2 and PA3. We now clarified this and lines 184-187 reads “Consistent with this, at E9.5 (20 somites) we found that the OFT was connected to PA2 and CNCCs from PA2 are entering the OFT (Fig. 2G, left). At late E9.5 (24 somites), the OFT was located between PA2 and PA3 and CNCCs from PA2 and PA3 were entering the OFT (Fig. 2G, right).”.

Figure 1: legend has no N numbers

The N numbers have been added.

Figure 1: Sox10 changes colour from red to blue to red again across Figure 1J-L which is confusing, can this be modified to be kept consistent?

We now modified this in the revised figure to be consistent.

Figures 2a and 3a: could the figure be modified to include better annotations for each cluster in the panel? It is difficult to look at the panel and have to refer back to the legend constantly.

Figures 2a and 3a have the same annotation as Figure 1. We added some more annotations.

Figure 2B: better choice of where to put the arrow heads could be made.

We have now more carefully placed arrowheads.

Figure 2C: I don't think some of those arrow heads are correct.... They don't seem to be pointing at the co-expressing cells?

We have now placed the arrowheads to the correct position.

Figure 2F,G: Can the authors clarify why Acta2 and Tagln antibodies are both used to label smooth muscle cells?

We used ACTA2 and TAGLN as markers of early smooth muscle cells. TAGLN is ubiquitously expressed in vascular smooth muscle cells and is an early marker of smooth muscle differentiation. We now included various smooth muscle markers to better characterize the cell types we consider as smooth muscle progenitor cells.

Reviewer #3 (Remarks to the Author):

De Bono and collaborators employ single-cell RNA sequencing (scRNA-seq) to (i) investigate developmental transitions that take place during crest cell (CNCC) development and (ii) examine the function of TBX transcription factors in this cell population.

The authors begin by characterizing the transcriptional profiles of Wnt1-Cre;EGFP positive cells from the pharyngeal region of E8.5, E9.5 and E10.5 embryos. They examine the transcriptional signature of the CNCC population at E9.5 and E10.5, which allows them to construct a map of CNCC fate that they validate through cell fate probability analysis using the CellRank pipeline. They subsequently perform scRNA-seq in wild-type and Tbx1 mutant embryos to identify differentially expressed genes. The results show that mutant embryos display upregulation of BMP genes and downregulation of FGF genes in the pharyngeal region. By analyzing data from Mesp1Cre control and Tbx1 conditional null embryos (at E9.5) with the CellChat software, the authors identify an increase of BMP signaling from the mesoderm to NCC in Tbx1 mutant embryos.

The datasets generated by the authors allow for the identification of a set of genes expressed during CNCC development. scRNA-seq of Tbx1 mutant embryos provides compelling evidence that loss of this TF results in the misregulation of the FGF and BMP signaling pathways. However, the study is limited by the strategy utilized to analyze the single cell data, which lacks precision. Instead of combining scRNA-seq datasets to delineate developmental trajectories (from 8dpc to 10.5dpc), data from each embryonic stage is analyzed separately. This limits insights into the developmental transitions that characterize CNCC development. In addition, the model proposed by the authors on the interplay between TBX transcription factors and FGF/BMP signaling is enticing but lacks further functional and biochemical validation. These shortcomings hinder the potential impact of the study.

We have now added the following data:

- 1) We now integrated the data from E8.5, E9.5 and E10.5 and the results are now shown in a new supplementary figure, Supplementary Figure 7. The validity of this integration is supported by similarity of neural tube cell populations at the three stages, as they do not change dramatically. Unlike for cells in the neural tube that are similar at all three stages, our data shows that pharyngeal NCCs at these three stages are quite distinct. Trajectory analysis of the integrated data provides supportive and confirmatory information but it doesn't provide significant new insights as compared to what we show in Figures 2 and 3.
- 2) We performed RNAscope *in situ* hybridization analysis to find the source of BMP signals. We examined expression of *Bmp4* and *Bmp7*. *Bmp4* is expressed in the second heart field mesoderm and *Bmp7* is expressed in the pharyngeal mesoderm and endoderm adjacent to NCCs. They continue to be robustly expressed in *Tbx1*^{-/-} embryos. We suggest that their presence along with

changes in BMP pathway genes are responsible for increased and premature expression of BMP downstream genes.

- 3) We tried to genetically rescue pharyngeal arch artery and/or OFT defects by normalizing BMP levels in *Tbx1*^{-/-} embryos. For this, we crossed in a *Smad1* floxed allele, but we did not have evidence for rescue (see details above for Reviewer 2). This lack of rescue may be explained because *Tbx1* also has functions in other tissues for PAA and OFT development; SMAD1 is expressed in NCCs in the neural tube and it might have independent functions, or that more than one type of signaling needs to be normalized for genetic rescue.
- 4) We tested for changes in MAPK signaling by performing immunofluorescence for p-ERK, and confirmed reduction of MAPK signaling.
- 5) We examined the literature for additional signaling downstream of *Tbx1*. We previously published on increased in canonical WNT signaling mediating OFT defects (PMID: 28346476). Evidence in our study is consistent with increased WNT signals from the second heart field, and this could affect NCCs function. We also added this to the model in Figure 8B.

Main concerns:

- The embryo dissections are arbitrary, which skews the proportions of cells that are being analyzed. The vast majority of the cells from 9.5dpc are cranial; PA1 gets dissected out in 10.5dpc, etc. While this does not compromise the analysis (as the authors can identify cNCCs), it does affect the quality of the datasets since the proportion of the populations of interest varies widely from stage to stage.

At E8.5, the embryo is too small to allow for a very precise dissection. At E8.5, we used the rostral half of the embryo that included the pharyngeal apparatus with heart. For E9.5, we removed the head because there are many NCCs that are not relevant to the cardiovascular system that we are studying and we didn't want to dilute out the pharyngeal NCCs including CNCCs. At E9.5 we included in the dissection, the pharyngeal apparatus from PA1 onwards and the heart, but removed the caudal half of the embryo. For E10.5, we excluded the head and PA1, because at this time point, these cells are destined to form the craniofacial skeleton, not the cardiovascular system. We were concerned that including the head and PA1 at E10.5, would dilute out the CNCC populations. We included the rest of the pharyngeal apparatus, PA2-6 and the heart, but excluded the caudal half of the embryo.

The rationale is better explained for this on page 631-638, and it reads, "The head of the embryo was conserved at E8.5 because of the small size of the embryos that most likely contain CNCC progenitors in the anterior part of the pharyngeal region. At E9.5, the pharyngeal region from PA1 onwards plus the heart were collected. The head of the embryos containing NCCs not relevant to the cardiovascular system were removed at E9.5 to not dilute out the pharyngeal NCCs including CNCCs. At E10.5, the pharyngeal region plus heart were collected. The head and PA1 were removed at E10.5 as shown in Fig. 1, because at this time point, these NCCs are destined to form the craniofacial skeleton, not the cardiovascular system."

The proportions in the two time points are not directly comparable and we did not compare the proportion of the populations between stages (E9.5 vs E10.5). We compared proportions of NCCs between controls and *Tbx1* null embryos from same developmental stage (E9.5 or E10.5) after scRNA-seq data integration as shown in Figure 5 A-D and Figure 7 A-D. Control and *Tbx1* null embryos at same stage were microdissected similarly.

- Re: Figure 1. - "...and data from 36,721 NCCs were obtained" – were these GFP+ cells or NCCs? *Wnt1-Cre* labels a lot of cells that are not NCCs. Also, where are the CNS cells in the UMAP? Were these filtered out? Are the cells labeled as "neuronal/neural" supposed to be crest-derived neural cells or CNS cells?

Thank you for your comment. We have now provided marker genes for each of the clusters of the *Wnt1-Cre* lineage cells at E8.5, E9.5 and E10.5 in Supplementary Figure 1, 2, 3. The *Wnt1-Cre* lineage labels a significant number of cells in the neural tube and are not just NCCs. We did remove (exclude) the head of the embryos containing the forebrain at E9.5 and E10.5, but we included the hindbrain and neural tube at the level of the pharyngeal apparatus. Neural cells refer to the cells deriving from the neural tube. Among them are neuronal cells that will form the cranial ganglia expressing genes like *Neurod1*. All the populations of NCCs that we analyzed are in Figure 1D-F.

- Re: Figure 1. - "Cluster C3 at E9.5 is similar to C2 at E10.5" –statement lacks explanation/quantification /statistics; it is not clear what the authors mean by this. Furthermore, the clustering assignment is based on a handful of genes and is not entirely convincing.

We removed this confusing statement relating C3 at E9.5 to C2 at E10.5 from the text.

We analyzed all the marker genes for each cluster and include this in Supplementary Figures 1, 2 and 3. Please see Figure 1 J, K and L as well as the Supplementary figures 1, 2 and 3 that has a subset of the marker genes.

Moreover, the authors have not included supplementary data on the control metrics that they used to determine the quality of their scRNA-seq dataset.

Supplementary Table 1 show the metrics of the quality of the scRNA-seq dataset that includes somites, cell viability, number of captured cells, mean reads/cell, median genes/cell and the regions dissected.

- The strategy to define cNCC gene signature in Fig 2 is confusing, as *Acta2* is only expressed in a very small number of cells.

In Figure 2 we identified the genes that are progressively activated in CNCC progenitors during their fate progression to smooth muscle cells. We used both *Acta2*, other markers of early smooth muscle cells (e.g. *Tagln*) and known transcription factors that function in NCCs (e.g. *Hand1/2*) for heart development. We better characterized early smooth muscle NCCs by performing additional immunostaining and RNAscope hybridization including *Cnn1*, *Rgs5*, *Tagln* (Figure 2 and 3). We refer to these cells as SM-CNCCs. This together, helped us to define the SM-CNCC gene signature, despite that there are not so many of these cells in the embryo at this time point. To identify fate progression, we used software to capture the cell fate transitions, in which cells differentiate to progenitors of smooth muscle cells, expressing *Acta2* and other markers, listed here and also provided in our heatmap. The PHATE map method captured the gene expression relationship between cells. We also used CellRank software because it combines gene expression similarity and RNA velocity, which can predict the differentiation trend of a cell. Therefore, when taken together, our data supports cell fate progression. And we also looked at the E10.5 data and saw an expansion of the SM-CNCC population and found similar patterns of cell fate progression. The E10.5 data with more SM-CNCCs validated the findings at E9.5 with fewer SM-CNCCs.

What about markers that have been shown to be crucial for cNCC identity (e.g. TGIF, MAFB, etc)?

We refer to these genes in the Introduction of the manuscript. Expression of *Sox8*, *Ets1* and *Tgif1* were identified in early migrating NCCs cells from the caudal rhombomeres that serve as progenitors of CNCCs in early chick embryos (Ghandi et al., PMID: 32369742). We examined expression of these genes and found that they weren't particularly useful as markers in our study of mouse embryos (please see UMAP plots below for controls at E8.5, 9.5 and E10.5). We found that *Sox8* and *Tgif1* were not specifically expressed in CNCCs at E8.5, E9.5 and E10.5. In addition, *Mafb* was indicated as an early marker of CNCCs. We found that *Mafb* was expressed in early migrating NCCs from PA3 at E8.5 and E9.5 consistent with the Gandhi et al. study. *Mafb* is also expressed at a low level in progenitor CNCCs expressing *Isl1*, *Gata3*, *Msx2*, *Hand2*, *Acta2* and *Tagln* and in some CNCCs progenitor expressing *Hoxb3*, *Tbx2* and *Tbx3* at E10.5. However, this gene was not found in our cardiac cell fate probabilities gene list suggesting that it is not a CNCC driver gene (Supplemental data 4 and 5). *Ets1*, is broadly expressed in pharyngeal NCCs at E8.5, E9.5 and E10.5 including early migrating NCCs in PA3, consistent with Gandhi et al. We found that *Ets1* shows early activation in pharyngeal NCCs and then a later staged activation when CNCCs start to express smooth muscle marker genes at E9.5 (Figure 2 K). *Ets1* was very weakly expressed at E10.5, but it is expressed early at E8.5 and E9.5. *Ets1* is as a potential CNCC driver gene as shown in the cell fate probabilities heatmaps in Figures 2 and 3 (supplementary data 5).

Additionally, the genes of the "CNCC genetic signature" (*Isl1*, *Gata*, *Hand*) have all been reported before, so I am unsure about much novelty was uncovered by the scRNAseq.

We have used *Isl1*, *Gata4-6*, *Hand1/2* as markers of transcription factors known to be important for cardiac development in a general sense, similarly we used *Acta2* and *Tagln* as markers of early smooth muscle cells. We included a subset of the new genes we identified in Figure 2J and K and 3N and O marked

by an asterisk. We also now include feature plots showing the gene expression patterns at E9.5 and E10.5 of the new genes identified (Supplementary Figure 6). The rest are present in the large heatmaps (Supplementary data 4 and 5). This includes *Tbx2/3* that we validated.

Figures 2 and 3 should be rearranged, as the text goes back and forth between them. Perhaps the results identifying and validating the CNCC gene signature for stages E9.5 and E10.5 (i.e., Figure 2A-G and Figure 3A-H) should be consolidated. Even better – the authors could combine the cNCCs of both stages (and perhaps include 8.5 dpc as well) to perform RNA velocity and inference of developmental trajectories. As it is, the datasets from different stages are underutilized.

We tried combining Figures 2 and 3 and this resulted in too many panels, and difficulty in the writing.

Following the Reviewer's helpful suggestion, we have performed integrated analysis of the cells from control embryos of E8.5, E9.5 and E10.5 (Supplementary Figure 7). The UMAP representation of the integrated data demonstrated that NCCs in the pharyngeal region in the three stages have very distinct gene expression profiles, while the cells remaining in the neural tube border are almost indistinguishable. As such, trajectory analysis based on RNA velocity alone (by scVelo) or RNA velocity + gene expression similarity (by CellRank) both indicate that no additional insights could be obtained from the integrated analysis. As such, we have decided to keep results from our original analysis of the E9.5 and E10.5 samples independently but include the new integrated results in Supplemental Figure 7. We do agree with the Reviewer that in most cases integration analysis, by spreading cells along a pseudotime trajectory, can provide insight into the genetic programs driving differentiation trajectory. In this case, the local environment likely plays a more important or dominant roles in determining the expression of the post-migrated cells, leading to little overlap of cells from the three stages.

The rationale for selecting the genes in Figure 3L as lineage driver genes whose expression correlates with SM-CNCC fate acquisition is unclear from the main text/methods. These genes represent a relatively small subset of the complete list of genes that show similar expression patterns when cells are ordered by smooth muscle fate probabilities (Supplementary data 5).

We agree that this was not spelled out properly. There are a total of 10 pages (>1000) of genes in the heatmap. This is provided in Supplementary data 5. Technically we needed to cut this down to be able to show a figure in the main text. Biologically, while all of the genes may be important for the lineage progression not all of them would play a driving role. We consider genes encoding transcription factors and signaling proteins are more likely to be the drivers. Thus, we selected genes in part, based upon the literature identifying them as being necessary for heart development and/or NCCs development but we also included some new genes we identified. We also performed gene ontology analysis of the complete dataset after dividing genes into groups by their pseudotimes (results of GO shown in Supplementary data 6). We have now spelled out our rationale for selecting the genes in the Results section (lines 209-211), "We mined the literature focusing on genes encoding transcription factors and signaling molecules, which traditionally have roles in cardiac development and modulating cell fate at E9.5 and E10.5 (Fig.2K and 3N)."

In the text (lines 219 – 221; re: Fig 5), the authors state that loss of *Tbx1* results in the increased expression of genes downstream of BMP signaling in clusters 4 and 8. However, they only show graphs summarizing differential gene expression and GO term analysis for cluster 8. What happens in cluster 4? PA3 (represented at E9.5 in cluster 4) is of particular importance since CNCCs will begin entering the OFT from this PA later during E9.5 (as the authors have previously shown).

We have updated Figure 5 and show that BMP signaling genes are also increased in cluster C4 together with reduced expressed of genes in the MAPK pathway (Figure 5F). We also provide violin plots showing *Msx2*, *Bambi*, *Spry1* and *Myc* expression, which replaced the GO networks that didn't provide further insights.

The rationale for focusing on BMP and MAPK pathway-related genes for the differential gene expression analysis centers around the importance of these signaling pathways for NCC development and migration during embryogenesis (as stated by the authors in lines 24-226). Yet, the gene ontology analysis of upregulated genes in cluster 8 also reveals the enrichment of genes involved in the regulation of other signaling pathways (e.g. Wnt), which are also consequential for NCCs development. This should be clarified.

We focused on BMP and MAPK because they were changed the most in our scRNA-seq dataset. We agree that there are several pathways that are altered as exemplified in Figure 6B (the bubble plot from CellChat; and Supplementary Figure 12-full bubble plot). We reported on increased WNT signaling before (PMID: 28346476) and now include this pathway in the model in Figure 8, as a representation of other pathways that are affected when *Tbx1* is inactivated. The BMP and MAPK were two that were most obviously changed among the differentially expressed genes and we validated them in this study.

The authors propose a model where mesodermal TBX1 promotes FGF signaling, which in turn inhibits BMP activity in CNCCs. The model is consistent with the results from the scRNA-seq analysis in the TBX1 mutants, but much of it seems speculative. For instance, I could not find data supporting that increase in BMP signaling in the mutants is due to the loss of FGF signaling. Epistatic experiments would be necessary to show that this is indeed the case (since other signaling pathways are also affected in the mutant). The study also lacks experiments that would show a direct interaction between the components of the model.

FGF signaling reduction shown in the model. There is extensive literature supporting the idea that FGF signaling is required downstream of *Tbx1*, both genetically and mechanistically. Inactivation of *Tbx1* results in reduced FGF8 and FGF10 signaling (PMID: 122234167; PMID: 17000704; PMID: 17238155). FGF ligands are well known to act downstream of *Tbx1* in the mesoderm and their loss indirectly results in CNCC mediated cardiovascular anomalies (PMID: 16720879; PMID: 21419761; PMID: 20035084). We now show that MAPK is reduced in CNCCs by examining p-ERK by immunostaining and the results supports the importance of FGF signaling from the mesoderm to MAPK in the NCCs (Figure 5).

BMP signaling increase shown in the model. There is strong published evidence that increase in BMP signaling affects NCC function and this leads to cardiac OFT defects. Inhibition of Dllard, a BMP antagonist increases BMP signaling and leads to OFT defects, which is consistent with the idea that increased BMP activity affects cardiac development similarly to that of *Tbx1* null mutants (PMID: 32105214). We found that BMP downstream genes are increased in expression and dorsally expanded. We found that pSMAD is increased and expressed in a dorsally expanded region in the pharyngeal apparatus, indicating premature activation and upregulation of BMP signaling when *Tbx1* is inactivated. We now performed expression studies of *Bmp4* and *Bmp7* and we found that both genes are strongly expressed in the pharyngeal apparatus in control and *Tbx1* null mutant embryos (Figure 6 and Supplementary Figure 13). This suggests that the source of the ligands is in neighboring cells that interact with NCCs. These ligands and altered expression of BMP pathway genes is likely the reason that BMP downstream genes are increased in expression.

The conclusion that reduced FGF leads to increased BMP. We speculated that loss in FGF signaling may result in an increase in BMP activity. This derived from a paper on antagonism between FGF and BMP pathways (PMID: 20847311). We did several experiments to examine this hypothesis but the main one was that we tried to rescue OFT defects in *Tbx1* null embryos by crossing in the *Smad1* floxed allele. We presented the results above (see for Reviewer 2). Unfortunately, we didn't observe genetic rescue but there are many reasons for this result and we provide some of the main ones above.

Other pathways are important downstream of *Tbx1*. We failed to include other pathways that are altered downstream of *Tbx1*, because our focus was on FGF and BMP, where we found the most changes. We previously identified upregulation of canonical WNT signaling when we inactivated *Tbx1* in the anterior second heart field mesoderm (PMID: 28346476). We identified that *Nrg1* expression was reduced when *Tbx1* was inactivated in the mesoderm (PMID: 34789765). Previously, it was shown that *Tbx1* acts upstream of *Wnt5a* in the formation of the cardiac OFT (PMID: 25410658). We also identified other ligand-receptor pairs that were altered. Validation of each one of these is a future goal, but we will not be able to do this in a short time. We included WNT signaling in addition to FGF-MAPK and BMP, just as another signaling pathway to be representative (Figure 8B). We conclude that many signaling pathways are perturbed in the mesoderm (and likely epithelia) upon *Tbx1* inactivation. We now have modified the model to broaden the important signaling and crosstalk that may act on CNCCs. We removed the causal relationship between reduced FGF and increased BMP from the model. This provides a more conservative and less speculative model that is more realistic with respect to the data we provide.

REVIEWERS' COMMENTS

Reviewer #1 (Remarks to the Author):

In the revised manuscript, De Bono and colleagues have attempted to answer each reviewer's concerns and comments. As a result, this manuscript is improved from the previous submission and is stronger in several aspects. However, there still remain concerns especially about the validity of cell fate transitions inferred from the present analysis (specific comment #3) and developmental implication of possible changes in BMP signaling (specific comment #6).

Although the PHATE mapping method is advantageous over other methods in that it well indicates the global structure of the dataset, the inference of fate transitions from cell fate probabilities based on snapshot data at each developmental stage is weak because of the intrinsic limitation of these techniques, and requires biological validation. As Reviewer #3 suggested, analysis combining the CNCC gene profiles of different stages would have somewhat improved this work. However, the authors state that such integration did not give additional insights. If they think that differences in the local environment may cause it, analysis of the combined data considering different special relationship might improve the outcome.

Regarding the implication of BMP signaling, it still remains unclear whether and how changes in BMP signaling are involved in the Tbx1-null phenotype in spite of a lot of additional efforts by the authors. In particular, it is reasonable that the phenotypic analysis of Tbx1 and Smad1double mutants is complex and equivocal given that BMP signaling has pleiotropic roles even in the same cell lineage. Although the authors provide possible explanations for their observed results, there remain concerns about the mechanistic involvement of BMP signaling within NCCs in the Tbx1-dependent genetic program.

Reviewer #2 (Remarks to the Author):

This is a detailed and comprehensive paper describing the transcriptome of cardiac neural crest cells that migrate through the pharyngeal region and to the outflow tract during a critical time of cardiovascular development in the mouse. The transcriptome of neural crest cells from Tbx1-null embryos has also been analysed and key gene expression differences identified.

The authors have responded to all comments to my satisfaction, and I recommend the paper to be accepted for publication.

Reviewer #3 (Remarks to the Author):

This is a much improved, extensively reviewed version of the manuscript by De Bono and colleagues, which I previously reviewed. Overall, I found the authors did a thorough job addressing my concerns, with the inclusion of a number of supplemental figures and quality control metrics.

One of my main issues with the study was that the scRNAseq analysis of the cardiac NCCs (more specifically the Wnt1-Cre-positive cells from the pharyngeal region) for each developmental stage (E8.5 -E10) was carried out separately, thereby potentially restricting what we can learn about the developmental transitions that take place between said stages.

To address this, the authors added a supplemental figure where they performed an integrated analysis of the three abovementioned stages, and found that the results mostly validated what was reported on Figures 2 and 3 (without, in their words, any significant added insight).

Another major concern was that the dissections to collect cardiac NCCs were arbitrary, which might have biased the proportion of NCCs used in their final analysis. The explanation provided by the authors on why they performed the dissections is reasonable. For the comparisons where maintaining the same proportion of cells really mattered, like their control vs Tbx1 null experiment, the embryos were dissected similarly at the respective stages.

Another concern was that "The strategy to define cNCC gene signature in Fig 2 is confusing, as Acta2 is only expressed in a very small number of cells." We were concerned about how the authors defined (or rather, did not clearly define) the gene signature of cardiac NCCs to identify these cells in the UMAP.

In their response, the authors explain that these ACTA-positive cells are indeed the cells that will become the smooth muscle cardiac NCCs, instead of clarifying their strategy to confirm those clusters are indeed cNCC. I found their explanation for why markers shown to be important for chick cardiac NCC identity are not useful for their analysis is acceptable. Most of it appears to be due to species-specific differences between mouse and chick (e.g., MAFB in mouse is faintly expressed in their cardiac NCC population and more highly expressed in early migrating NCCs).

The authors also refined the proposed model, which is now better integrated with the existing literature.

Response to the Reviewers. We thank the three reviewers for their insightful comments. We include a point by point response to the reviewer's comments in blue font.

REVIEWERS' COMMENTS

Reviewer #1 (Remarks to the Author):

In the revised manuscript, De Bono and colleagues have attempted to answer each reviewer's concerns and comments. As a result, this manuscript is improved from the previous submission and is stronger in several aspects. However, there still remain concerns especially about the validity of cell fate transitions inferred from the present analysis (specific comment #3) and developmental implication of possible changes in BMP signaling (specific comment #6).

Although the PHATE mapping method is advantageous over other methods in that it well indicates the global structure of the dataset, the inference of fate transitions from cell fate probabilities based on snapshot data at each developmental stage is weak because of the intrinsic limitation of these techniques, and requires biological validation. As Reviewer #3 suggested, analysis combining the CNCC gene profiles of different stages would have somewhat improved this work. However, the authors state that such integration did not give additional insights. If they think that differences in the local environment may cause it, analysis of the combined data considering different special relationship might improve the outcome.

Important morphological and transcriptomic changes occur in murine embryos between E8.5, E9.5 and E10.5. This is reflected in our integration analysis of scRNA-seq data at E8.5, E9.5 and E10.5, where we failed to observe significant overlap in cell populations at different stages. We agree with the Reviewer and suggest that transcriptomic changes in NCCs could be due to changes in their microenvironment over developmental time. For example, the pharyngeal arches form in a segmented manner and there are distinct differences in the number of arches at the three stages, in which all arches are formed by E10.5. Changes in local environment can be more fully investigated on a single cell level by utilization of spatial transcriptomics technology. This technology is only in its infancy at the moment but as it improves, we may uncover novel insights about cell transcriptomics in different positions in the embryo at different stages.

Regarding the implication of BMP signaling, it still remains unclear whether and how changes in BMP signaling are involved in the Tbx1-null phenotype in spite of a lot of additional efforts by the authors. In particular, it is reasonable that the phenotypic analysis of Tbx1 and Smad1double mutants is complex and equivocal given that BMP signaling has pleiotropic roles even in the same cell lineage. Although the authors provide possible explanations for their observed results, there remain concerns about the mechanistic involvement of BMP signaling within NCCs in the Tbx1-dependent genetic program.

We agree with the statement and suggest that in the future, it will be necessary to perform in vitro studies in cell culture. For example in investigating organoids or differentiating embryonic stem cells in cell culture, while exposing cells of particular lineages to morphogens and

antagonists. This might help resolve this question as part of future investigation. We state the following in the Discussion: “In the future utilization of organoids and cell culture systems will help dissect the mechanism of how BMP interacts with NCCs.”

Reviewer #2 (Remarks to the Author):

This is a detailed and comprehensive paper describing the transcriptome of cardiac neural crest cells that migrate through the pharyngeal region and to the outflow tract during a critical time of cardiovascular development in the mouse. The transcriptome of neural crest cells from *Tbx1*-null embryos has also been analysed and key gene expression differences identified.

The authors have responded to all comments to my satisfaction, and I recommend the paper to be accepted for publication.

Reviewer #3 (Remarks to the Author):

This is a much improved, extensively reviewed version of the manuscript by De Bono and colleagues, which I previously reviewed. Overall, I found the authors did a thorough job addressing my concerns, with the inclusion of a number of supplemental figures and quality control metrics.

One of my main issues with the study was that the scRNAseq analysis of the cardiac NCCs (more specifically the *Wnt1*-Cre-positive cells from the pharyngeal region) for each developmental stage (E8.5 -E10) was carried out separately, thereby potentially restricting what we can learn about the developmental transitions that take place between said stages. To address this, the authors added a supplemental figure where they performed an integrated analysis of the three above mentioned stages, and found that the results mostly validated what was reported on Figures 2 and 3 (without, in their words, any significant added insight).

We integrated our scRNA-seq datasets from E8.5, E9.5 and E10.5 embryos. Our integration analysis revealed that the transcriptional profiles of the pharyngeal NCCs are quite distinct between the three stages, with the majority of E9.5 and E10.5 pharyngeal NCCs not clustered together in Supplementary Fig. 7a. However our analysis of cell fate probabilities confirms the progression from pharyngeal NCCs to cardiac NCCs identified in individual scRNA-seq data analyses of E9.5 and E10.5. Furthermore, progressive activation of some genes including *Msx2*, *Tbx2*, *Tbx3*, *Hand1*, *Hand2*, *Isl1*, *Gata3* and *Acta2* during the transition from pharyngeal NCCs to smooth muscle NCCs have been found in our integrated scRNA-seq data analysis.

Another major concern was that the dissections to collected cardiac NCCs were arbitrary, which might have biased the proportion of NCCs used in their final analysis. The explanation provided by the authors on why they performed the dissections is reasonable. For the comparisons where

maintaining the same proportion of cells really mattered, like their control vs Tbx1 null experiment, the embryos were dissected similarly at the respective stages.

Another concern was that "The strategy to define cNCC gene signature in Fig 2 is confusing, as *Acta2* is only expressed in a very small number of cells." We were concerned about how the authors defined (or rather, did not clearly define) the gene signature of cardiac NCCs to identify these cells in the UMAP. In their response, the authors explain that these ACTA-positive cells are indeed the cells that will become the smooth muscle cardiac NCCs, instead of clarifying their strategy to confirm those clusters are indeed cNCC.

ACTA2 is one of the cardiac smooth muscle markers. In order to confirm that ACTA2 positive NCCs are indeed cardiac NCCs at E9.5 (Fig. 2), we included immunostaining for TAGLN that is another early marker of cardiac smooth muscle cells. In addition, we investigated the list of marker genes (Supplementary data 2) of cluster C14 in Fig 2a that corresponds to cardiac smooth muscle NCCs. The list include genes such as *Acta2*, *Tagln*, *Rgs5*, *Cnn1*, *Myh9* and *Myh9* that are early markers of cardiac smooth muscle cells. This allowed us to define cells in cluster 14 as cardiac smooth muscle cells. Then we found that several genes important for cardiac development, including *Tbx2*, *Tbx3*, *Isl1*, *Msx2*, *Gata3* and *Hand1* expressed in cluster C14 are also expressed in cluster C1 and C3. This suggested that clusters C1 and C3 in Fig 2a contain cardiac NCC progenitors. We confirmed the expression of some of these genes (*Gata3*, *Isl1* and *Msx2*: Fig 2d-e) in NCCs the distal OFT and in the mesenchyme located dorsally to the OFT. Finally, our bioinformatic cell fate probabilities indicate that cells in clusters C1 and C3 have the potential to differentiate into smooth muscle cells of cluster C14 (Fig. 2h,i,k). Taken together, this indicates that clusters C1 and C3 contain cardiac NCC progenitors. This is explained in the text in the paragraph in the Results section entitled, "**Identification of cardiac NCC gene signatures**"

I found their explanation for why markers shown to be important for chick cardiac NCC identity are not useful for their analysis is acceptable. Most of it appears to be due to species-specific differences between mouse and chick (e.g., MAFB in mouse is faintly expressed in their cardiac NCC population and more highly expressed in early migrating NCCs).

The authors also refined the proposed model, which is now better integrated with the existing literature.